# Ly6G+ granulocytes-derived IL-17 limits protective host responses and promotes tuberculosis pathogenesis

Priya Sharma[1], Raman Deep Sharma[1], Binayak Sarkar[2], Varnika Panwar[1], Mrinmoy Das[1], Lakshya Veer Singh[1], Neharika Jain[1], Shivam Chaturvedi[1], Lalita Mehra[3], Aditya Rathee[1], Shilpa Sharma[1], Shihui Foo[4], Andrea Lee[5], Pavan Kumar N[6], Prasenjit Das[3], Vijay Viswanathan[7], Hardy Kornfeld[8], Shanshan W Howland[4], Subash Babu[9,10], Vinay Kumar Nandicoori[2,11], Amit Singhal[5,12], Dhiraj Kumar[1]*

[1]Cellular Immunology Group, International Centre for Genetic Engineering and Biotechnology, New Delhi, India; [2]National Institute of Immunology, New Delhi, India; [3]Department of Pathology, All India Institute of Medical Sciences, New Delhi, India; [4]Singapore Immunology Network (SIgN), Agency for Science, Technology and Research (A*STAR), Singapore, Singapore; [5]Infectious Diseases Labs (ID labs), Agency for Science, Technology and Research (A*STAR), Singapore, Singapore; [6]Department of Immunology, ICMR-National Institute for Research in Tuberculosis, Chennai, India; [7]Prof. M. Viswanathan Diabetes Research Center, Chennai, India; [8]Department of Medicine, University of Massachusetts Medical School, Worcester, United States; [9]NIH-International Center of Excellence in Research, Chennai, India; [10]Laboratory of Parasitic Diseases, NIAID-NIH, Bethesda, United States; [11]Centre for Cellular and Molecular Biology, Hyderabad, India; [12]Lee Kong Chian School of Medicine, Nanyang Technological University (NTU), Singapore, Singapore

*For correspondence: dhiraj@icgeb.res.in

## eLife Assessment

This **valuable** study examines the role of IL17-producing Ly6G PMNs as a reservoir for *Mycobacterium tuberculosis* to evade host killing activated by BCG immunisation. The authors provide **solid** data reporting that IL17-producing polymorphonuclear neutrophils harbour a significant bacterial load in both wild-type and IFNg-/- mice and that targeting IL17 and Cox2 improved disease outcomes whilst enhancing BCG efficacy. The specific contribution of neutrophil-derived IL-17 to disease pathogenesis remains to be definitively established through direct demonstration of IL-17 production by neutrophils and targeted depletion studies.

**Abstract** The protective correlates of *Mycobacterium tuberculosis* (*Mtb*) infection-elicited host immune responses are incompletely understood. Here, we report pro-pathogenic crosstalk involving Ly6G+ granulocytes (Ly6G+Gra), IL-17, and COX2. We show that in the lungs of *Mtb*-infected wild-type mice, either BCG-vaccinated or not, most intracellular bacilli are Ly6G+Gra-resident 4 weeks post-infection onwards. In the genetically susceptible *ifng*-/- mice, excessive Ly6G+Gra infiltration correlates with severe bacteremia. Neutralizing IL-17 (anti-IL17mAb) and COX2 inhibition by celecoxib reverse Ly6G+Gra infiltration, associated pathology, and death in *ifng*-/- mice. Surprisingly, Ly6G+Gra also serves as the major source of IL-17 in the lungs of *Mtb*-infected WT or *ifng*-/- mice. The IL-17-COX2-Ly6G+Gra interplay also operates in WT mice. Inhibiting RORγt, the key transcription factor for IL-17 production or COX2, reduces the bacterial burden in Ly6G+Gra, leading to

reduced bacterial burden and pathology in the lungs of WT mice. In the *Mtb*-infected WT mice, COX2 inhibition abrogates IL-17 levels in the lung homogenates and significantly enhances BCG's protective efficacy, mainly by targeting the Ly6G+Gra-resident *Mtb* pool, a phenotype also observed when IL-17 is blocked by RORγt inhibitor. Furthermore, in pulmonary TB patients, high neutrophil count and IL-17 correlated with adverse treatment outcomes. Together, our results suggest that IL-17 and PGE2 are the negative correlates of protection, and we propose targeting the pro-pathogenic IL-17-COX2-Ly6G+Gra axis for TB prevention and therapy.

## Introduction

Tuberculosis (TB) is the major cause of mortality and morbidity due to an infectious disease. A lack of an effective vaccine, prognostic diversity among the patients, and the emergence of drug resistance greatly hamper the efforts to control TB. BCG, the only available vaccine, has limited efficacy in adults, although it works well in preventing childhood TB and TB meningitis. A comprehensive understanding of how the host immune system responds to vaccination, infection, and treatment is fundamental to the success of TB elimination programs (*Chandra et al., 2022*).

Individuals exposed to *Mtb* follow a varied trajectory on parameters like developing latent or active disease, disease severity, and responsiveness to anti-TB chemotherapy (*Barry et al., 2009*; *Cadena et al., 2017*; *Dartois and Rubin, 2022*; *Heyckendorf et al., 2021*; *Pai et al., 2016*; *Thompson et al., 2017*). For example, the average treatment success rate for TB is ~80%, which is further lower in the case of drug-resistant TB (DR-TB, 50%) (*Chaves Torres et al., 2019*; *Rockwood et al., 2016*). Immunological features that contribute to treatment success and failure in TB patients present a unique opportunity to understand the host immunity associated with protection and pathogenesis against TB, respectively. The host-driven immunological mechanisms also play a critical role in the differential BCG responses, for example, contrasting T cell responses seen among young children and adults (*Brazier and McShane, 2020*), which fade over time (*Soares et al., 2013*). For intracellular pathogens like *Mtb*, the cell-mediated adaptive immune arm has been extensively explored for studying vaccine efficacy (*Brazier and McShane, 2020*). However, since the bacteria mostly reside inside the innate immune cells, it is pertinent to ask whether innate immune-based mechanisms also contribute to the differential vaccine-induced protective responses.

A variety of innate immune cells are present in TB granulomas and play a role in pathogenesis and control. For example, in addition to macrophages, TB granulomas also comprise other innate immune cells, like granulocytes, dendritic cells, and unconventional cells like mesenchymal stem cells (MSCs), bone cells, adipocytes, etc (*Cadena et al., 2017*; *Das et al., 2013*; *Fatima et al., 2020*; *Jain et al., 2020*). Whether different cellular constituents confound the protective host responses remains poorly understood. The bacteria residing in the MSCs can escape immune cytokines like IFNγ and TNFα, which could effectively blunt the T-cell-mediated protective immune responses (*Jain et al., 2020*). Similarly, neutrophils, the most extensively studied granulocytes in the context of TB (*Dallenga and Schaible, 2016*; *Kimmey et al., 2015*; *Lovewell et al., 2021*; *Mishra et al., 2017*) represent the most common *Mtb*-containing innate cells in the airways of active TB patients (*Eum et al., 2010*). Studies over the past decade point to a more pathological role of neutrophils in TB (*Berry et al., 2010*; *Han et al., 2018*; *Lowe et al., 2013*; *Nwongbouwoh Muefong et al., 2022*; *Mayer-Barber, 2023*), although some studies also report their protective roles in specific contexts (*Lowe et al., 2018*; *Martineau et al., 2007*; *Silva et al., 1989*). Neutrophils are a diverse cell type; however, Ly6G+ (both high and mid) granulocytes are the most studied subtype.

Using the mice model of TB, in this study, we explored neutrophils as a distinctive niche for *Mtb*. We report that Ly6G+ neutrophils (Ly6+Gra) infiltration and resident *Mtb* within these cells correlate with disease severity. We show the establishment of an IL-17-Ly6G+Gra axis, supported by COX2 in disease pathology. Our results indicate that disrupting the COX2-IL17-Ly6G+Gra axis significantly enhances the preventive responses of BCG, enables better control in genetically vulnerable subjects, and provides an exciting opportunity for host-directed therapy against TB.

## Results

### Neutrophils serve as the major niche for *Mycobacterium tuberculosis* in mice lungs

To assess diverse cellular niches in the lungs of *Mtb*-infected C57BL/6 (WT) mice, we investigated intracellular *Mtb* burden in Ly6G⁺ CD11b⁺ granulocytes (or Ly6G⁺Gra or PMNs), CD64⁺MerTK⁺ macrophages and Sca1⁺CD90.1⁺CD73⁺ mesenchymal stem cells (MSCs). The gating strategy for sorting these cells is shown in *Figure 1A*. Mice were infected with 100–200 H37Rv colony-forming units (CFUs) via aerosol challenge, and at 4, 8, and, 12 weeks post-infection (p.i.), the total bacterial burden in the lung and spleen was estimated by CFU enumeration (*Figure 1—figure supplement 1A–C*). In addition, from the lungs, we flow-sorted Ly6G⁺Gra, macrophages, and MSCs following the gating strategy shown in *Figure 1A*. Among the three cell populations, MSCs were the most abundant cell type across each of the time points post-*Mtb* infection (2–6x10$^4$/million sorted cells/half lung), followed by macrophages (4–9x10$^3$/million sorted cells) and Ly6G⁺Gra (1–4x10$^3$ / million sorted cells, *Figure 1—figure supplement 1D*). However, Ly6G⁺Gra contained, on average, four to five times more total bacillary load (from the total sorted pool of cells) as compared to either macrophages or MSCs (*Figure 1—figure supplement 1E*). To account for the differences in the total cell number obtained for the three cell types, we calculated CFU/10$^4$ cells, which showed the highest bacillary load in the Ly6G⁺Gra (*Figure 1B*). Tissue sections from the lungs of *Mtb*-infected mice also showed co-staining of Ly6G with Ag85B, the *Mtb* surface antigen (*Figure 1C*). Thus, our data suggest that a large number of *Mtb* in the lungs of infected animals reside inside the Ly6G⁺Gra, which outnumber those present in macrophages or MSCs.

### The residual *Mtb* population in BCG-vaccinated animals is largely Ly6G⁺Gra- or PMN-resident

To understand the relevance of Ly6G⁺Gra-resident *Mtb* and its contribution towards TB pathogenesis, we repeated the above experiments in BCG-vaccinated animals. Mice were administered either PBS or BCG (Pasteur strain) intradermally, followed by aerosol challenge with H37Rv at 8 weeks post-BCG vaccination (*Figure 1D*). As expected, animals that received BCG showed nearly 10-fold lower bacterial burden in the lungs and spleen compared to the unvaccinated PBS controls (*Figure 1—figure supplement 1F*). Consequently, BCG-vaccinated animals also showed less pathology than the unvaccinated PBS controls (*Figure 1—figure supplement 1G*). Among the sorted cells from the lungs, except for a marginal decline in Ly6G⁺Gra number at 8 weeks p.i. in the BCG-vaccinated group, the relative distribution of PMNs, macrophages, and MSCs in both control and BCG-vaccinated animals remained unchanged over the course of infection (*Figure 1—figure supplement 1H*). The *Mtb* burden, however, in each of the three cell types dropped significantly (by 80–90%) in the BCG-vaccinated animals compared to the unvaccinated animals (*Figure 1E*). Interestingly, lung Ly6G⁺Gra in BCG-vaccinated mice still contained 10–20-fold (~$\log_{10}$ 1-1.5 fold) more bacilli compared to the other two cell types for each time point p.i. investigated (*Figure 1E*). These results indicate that BCG-induced protective host immunity, despite being effective, was not sufficient to clear the Ly6G⁺Gra-resident *Mtb* population, unlike the *Mtb* pools present in MSCs or macrophages. Thus, we inferred that Ly6G⁺Gra serve as a preferred niche for *Mtb* to withstand host immunological assaults in mice.

### RNAseq analysis of Ly6G⁺Gra sorted from the lungs of *Mtb*-infected animals

We next analyzed the global gene expression profile of flow-sorted Ly6G⁺Gra from the lungs of unvaccinated or BCG-vaccinated mice, either uninfected or *Mtb*-infected (*Figure 1F*, *Supplementary file 1*). Only 27 differentially expressed genes (DEGs) were found between unvaccinated, uninfected (UI) and BCG-vaccinated, uninfected animals, and therefore, the BCG-vaccinated, uninfected mice group was excluded from the further analysis (*Supplementary file 2*). Compared to the UI control group, 944 genes were DEGs in the *Mtb*-infection group alone (Rv, blue circle), whereas there were 620 DEGs in the BCG-vaccinated *Mtb*-infected group (red circle), 373 genes being common to both the lists (*Figure 1F*). Compared to the *Mtb*-infected group, the BCG-vaccinated *Mtb*-infected group had 209 DEGs (green circle, *Figure 1F*). A gene set co-regulation analysis (GESECA, see methods) among DEGs across the three groups showed enrichment of Gene Ontology Biological Processes (GO-BP)

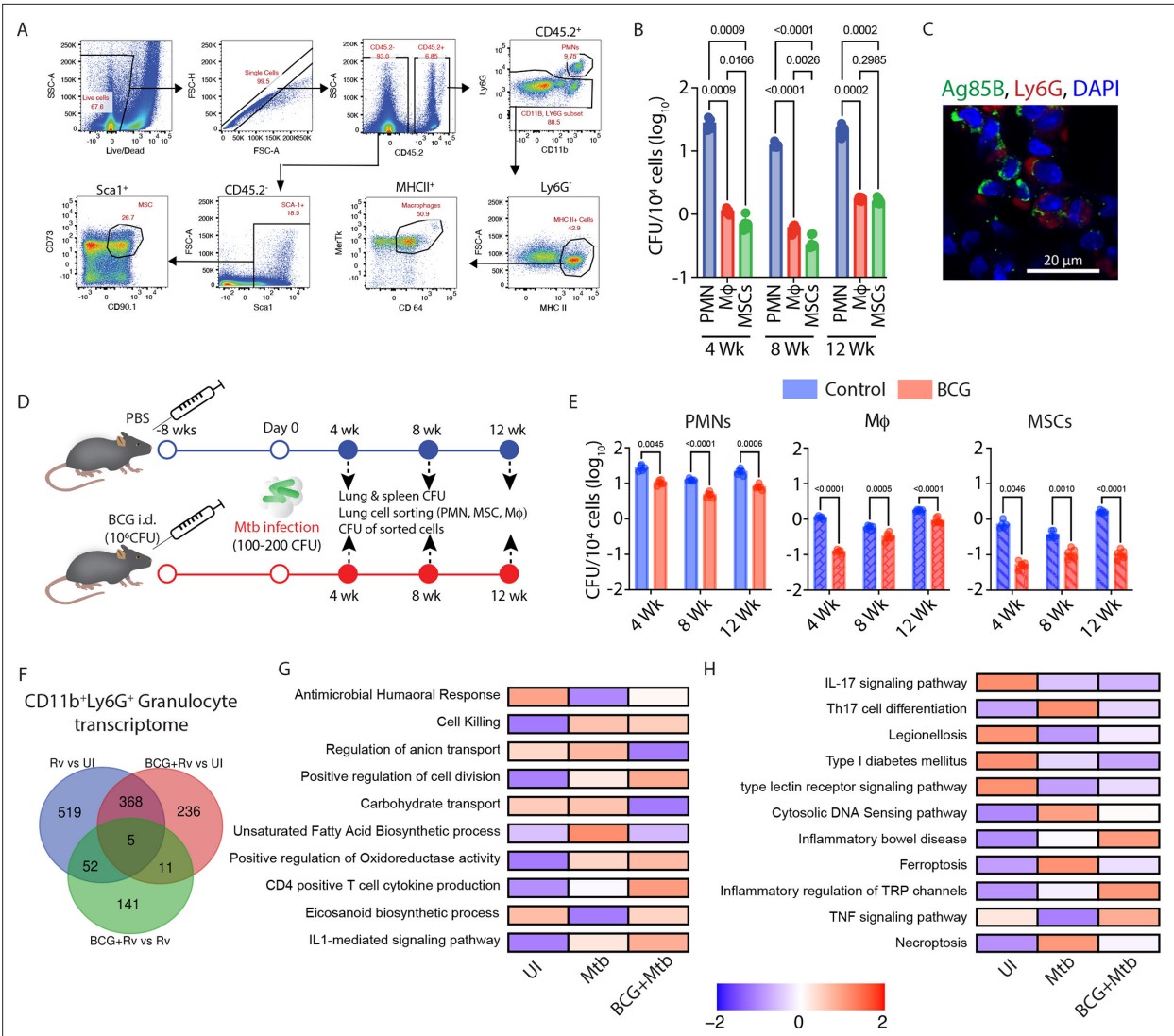

**Figure 1.** PMNs harbors substantial amounts of *Mtb* and provide an immune-privileged niche to *Mtb*. (**A**) FACS gating strategy used to sort PMNs, macrophages, and MSCs from the lungs of mice. (**B**) *Mtb* burden in individual sorted cells (PMNs, macrophages, and MSCs), plotted as CFU/$10^4$ sorted cells from the lungs of *Mtb*-infected mice at 4-, 8-, and 12 weeks post-infection time point. A two-way ANOVA statistical test was applied to compare all the groups at different time points. Sample size (n)=5 mice at each time point. (**C**) IFA images of the lungs of *Mtb*-infected mice showing the co-localization of PMNs (Ly6G$^+$ cells shown in red color) and *Mtb* antigen Ag85B (shown in green color). (**D**) Study design for BCG vaccination experiment. C57BL/6 mice were vaccinated intradermally with $10^6$ BCG bacilli, followed by aerosol infection with H37Rv (100–200 CFU) 8 weeks post-BCG vaccination. At 4, 8, and 12 weeks post-infection, mice were euthanized for various studies. (**E**) Determination of *Mtb* burden in individual sorted cells, plotted as CFU/$10^4$ sorted cells at 4, 8, and 12 weeks post-infection, from the lungs of vaccinated and unvaccinated mice. Experimental groups were evaluated using a two-way ANOVA statistical test. Sample size (n)=5 mice/group in all treatment groups. This BCG vaccination experiment was carried out independently in two distinct experimental replicates. (**F**) Venn diagram highlighting the number of common and unique differentially expressed genes (DEGs) from each pairwise comparison between BCG vaccinated +infected, only infected, and uninfected controls (**G**) z-scaled expression of the DEGs corresponding to the significantly enriched Gene Ontology Biological processes obtained using gene set co-regulation analysis of the DEGs (**H**) highlights the enriched KEGG pathways based on gene set co-regulation analysis of the DEGs. Within the plotGesecaTable() function, z-scores are first calculated for each normalized gene expression across the dataset. For each GO BP/KEGG pathway, the expression of its corresponding genes are summed up per sample and scaled and defined within the range of –2 to 2.

The online version of this article includes the following figure supplement(s) for figure 1:

**Figure supplement 1.** PMNs harbor a substantial amount of *Mtb*.

unsaturated fatty-acid biosynthesis in *Mtb*-infected group alone (*Figure 1G*). This included genes like *Sirt1*, *FadS3*, *IL1b*, *Scd1*, *Scd2*, and *Alox15* (*Supplementary file 3*). In the BCG-vaccinated and *Mtb*-infected group (BCG +*Mtb*), GO-BPs such as positive regulation of cell division, CD4[+]T cell cytokine production, and eicosanoid biosynthetic process were upregulated (*Figure 1G*). We also investigated the enrichment of KEGG pathways in DEGs and found Th17 cell differentiation, cytosolic DNA sensing pathway, ferroptosis, and necroptosis pathways to be upregulated in the *Mtb*-infected group but downregulated in the BCG +*Mtb* group (*Figure 1H*, *Supplementary file 4*). Th17 cell chemotaxis has been associated with exacerbated TB pathology, and therefore, a decline in this pathway in the BCG-vaccinated group concurred with earlier studies (*Jung et al., 2022*; *Basile et al., 2011*; *Jurado et al., 2012*; *Mills, 2023*). Together, the results from RNAseq analysis reveal the potential roles of the eicosanoid and IL-17 signaling pathways in regulating the Ly6G[+]Gra-mediated effects in TB.

## Association of Ly6G[+]Gra with severe TB pathology in *ifng[-/-]* mice

The results so far demonstrate a possible role of Ly6G[+]Gra-resident *Mtb* in TB pathology. To further investigate the mechanism of Ly6G[+]Gra-driven pathology, we used *ifng[-/-]* animals, which are known to develop severe neutrophilia and pathology upon *Mtb* infection (*Cooper et al., 1993*; *Flynn et al., 1993*; *Pearl et al., 2001*). We first verified the disease severity in *Mtb*-infected WT or *ifng[-/-]* mice (*Figure 2A*). The *ifng[-/-]* mice, compared to the WT mice, showed severe gross lung pathology (*Figure 2—figure supplement 1A*) and a very high *Mtb* burden in the lungs and spleen (*Figure 2—figure supplement 1B*), compared to WT mice, which was associated with high mortality (*Figure 2—figure supplement 1C*). Histopathology analysis revealed massive infiltration of the immune cells in the lungs of *Mtb*-infected *ifng[-/-]* mice (*Figure 2B*). Immunofluorescence staining of lung sections and flow cytometric analysis of the lung cells showed that most of the lung immune cells in *Mtb*-infected *ifng[-/-]* mice were Ly6G[+]Gra (*Figure 2C–D*). Unlike in the *Mtb*-infected *WT* mice, Ly6G[+]Gra recruitment followed a time-dependent increase in the lungs of *Mtb*-infected *ifng[-/-]* mice, increasing from ~26% at four weeks to ~61% at 10 weeks p.i. (*Figure 2D*). This increase was also reflected in the total Ly6G[+]Gra sorted from the lungs of these mice (*Figure 2—figure supplement 1D*). On average, *Mtb*-infected *ifng[-/-]* mice showed a nearly 40-fold higher number of Ly6G[+]Gra than the *Mtb*-infected WT animals (*Figure 2—figure supplement 1D*). Moreover, Ly6G[+]Gra-resident *Mtb* burden also increased several folds in the *ifng[-/-]* compared to the WT mice at 10 weeks p.i. (*Figure 2E*). Thus, massive infiltration of Ly6G[+]Gra and a very high intracellular *Mtb* burden in those cells correlated with severe pathology in *ifng[-/-]* mice.

## Severe TB pathology in *ifng[-/-]* mice correlated with increased lung IL-17 levels

To understand what might have resulted in the increased Ly6G[+]Gra infiltration in the *Mtb*-infected *ifng[-/-]* animals, we analyzed cytokine/chemokine levels in the lung homogenates. Several cytokines/chemokines were significantly dysregulated in the *Mtb*-infected *ifng[-/-]* mice compared to the *Mtb*-infected WT mice (*Figure 2F–G*, *Figure 2—figure supplement 1E*). Among those, CXCL1, CXCL2, IL-1β, IL-1α, IL-6, IL-17A (or IL-17), M-CSF, CCL3, and TNFα were significantly upregulated in the *Mtb*-infected *ifng[-/-]* compared to the *Mtb*-infected WT mice (*Figure 2F*). On the contrary, IFNγ, IL-12, IL-9, IL-10, IL-15, IL-18, IL-27, and IL-28 were downregulated in the *Mtb*-infected *ifng[-/-]* mice (*Figure 2G*). Among the upregulated cytokines in *Mtb*-infected *ifng[-/-]* animals, CXCL1 and CXCL2 are the known chemo-attractants for PMNs recruitment (*Kolaczkowska and Kubes, 2013*), whereas IL-1α, IL-1β, and IL-6 are key players in the differentiation and expansion of the Th17 cells (*Ketelut-Carneiro et al., 2019*; *Lee et al., 2010*; *Mills, 2023*). Interestingly, IL-17, a classical pro-inflammatory cytokine predominantly secreted by the Th17 cells, was also significantly upregulated in the *Mtb*-infected *ifng[-/-]* animals (*Figure 2F*). Increased IL-17 in the *Mtb*-infected *ifng[-/-]* mice concurred with earlier reports which show a key role for IL-17 in TB disease pathology (*Cruz et al., 2010*).

## Ly6G[+]Gra are the key source of IL-17 in TB granulomas

IL-17 is important for TB pathology (*Mills, 2023*), and also for the recruitment and activation of neutrophils (*Allen et al., 2015*). For Th17 cell effector function and to sustain a high level of IL-17 production, another cytokine, IL-23, is required (*Haines et al., 2013*; *Langrish et al., 2005*; *Teng et al., 2015*). However, in our cytokine analysis, we did not observe any increase in IL-23 production in

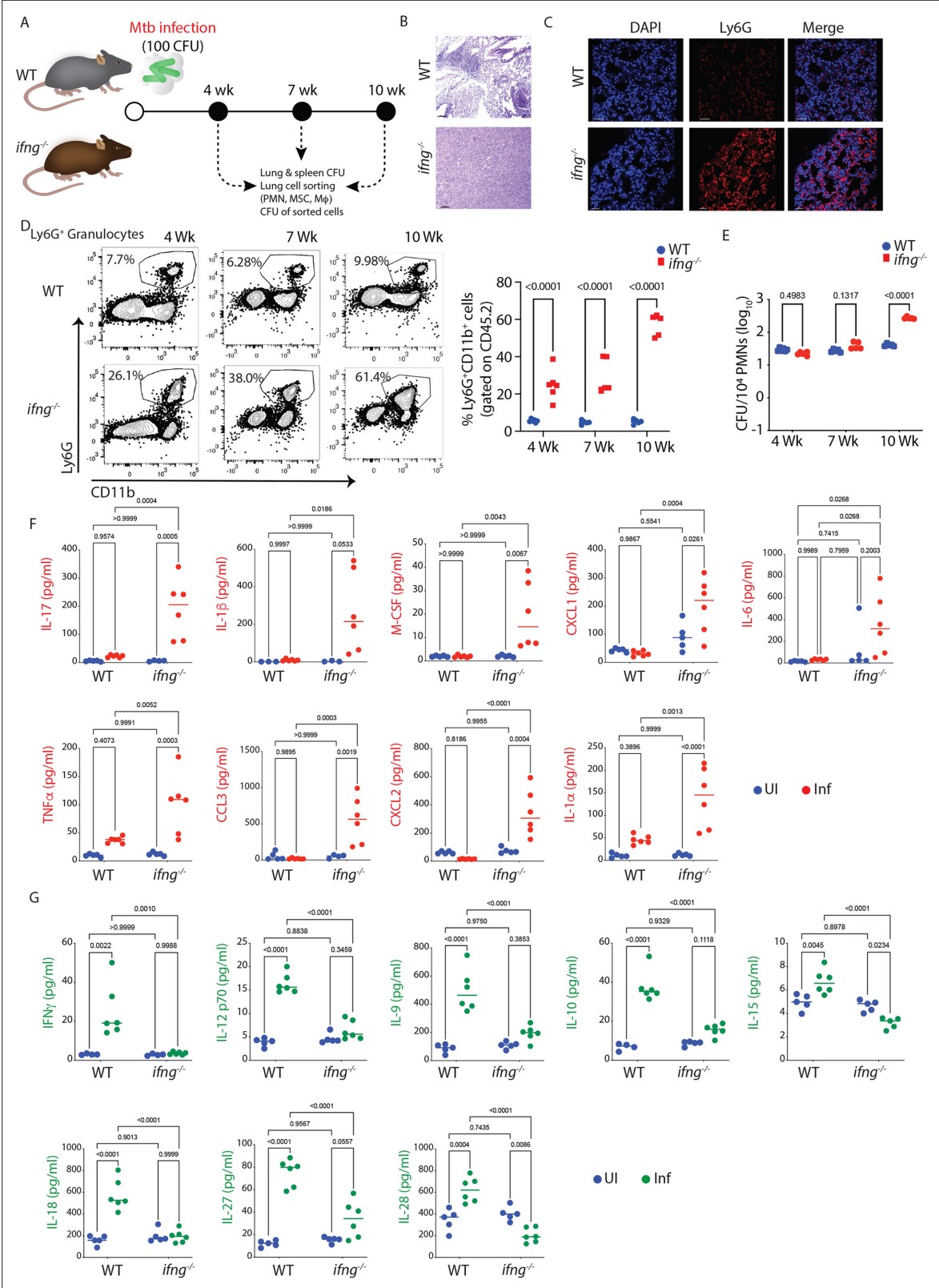

**Figure 2.** Uncontrolled neutrophilia, huge *Mtb* burden, severe disease pathology, and death in *ifng*⁻/⁻ mice is mediated by high IL-17. (**A**) Experimental design and timeline. C57BL/6 mice (WT and *ifng*⁻/⁻) were infected with H37Rv, 100 CFU, via the aerosol route. Mice from both groups were euthanized at 4, 7, and 10 weeks post-infection time point. Lung and spleen tissues were harvested and processed for diverse assays, as shown in the study design. (**B**) H&E-stained lung sections of WT and *ifng*⁻/⁻ mice at ×10 magnification, showing severe histopathology in the latter group of mice. (**C**) Lung IFA

*Figure 2 continued on next page*

*Figure 2 continued*

images from the same group of mice showing the extensive infiltration of PMNs (Ly6G+ cells stained in red color) in *Mtb*-infected *ifng*-/- mice compared to WT mice. (**D**) Representative FACS dot plot showing the percentage of PMNs in the lungs of *Mtb*-infected WT and *ifng*-/- mice (population is gated on CD45+ cells). Percentage of PMNs in the lungs of *Mtb*-infected WT and *ifng*-/- mice (population is gated on CD45+ cells) is shown at the right. (**E**) *Mtb* burden in the PMNs, sorted from the lung of *Mtb*-infected WT and *ifng*-/-, plotted as CFU/10$^4$ sorted cells at 4, 7, and 10 weeks post-infection. Experimental groups were evaluated using a two-way ANOVA statistical test. Sample size (n)=5–6 mice/group in both the groups. Luminex data: Quantification of several cytokines and chemokines from the lung supernatant of uninfected and *Mtb*-infected WT and *ifng*-/- mice. (**F**) Cytokines and chemokines that are upregulated in *Mtb*-infected *ifng*-/- mice compared to WT mice (**G**) Cytokines and chemokines that are downregulated in *Mtb*-infected *ifng*-/- mice compared to WT mice. Data were evaluated using a two-way ANOVA statistical test. Sample size (n)=5–6 mice/group in all the groups. This experiment was performed only once as a single experimental replicate.

The online version of this article includes the following figure supplement(s) for figure 2:

**Figure supplement 1.** Contribution of neutrophilia to the severity of TB disease.

*Mtb*-infected *ifng*-/- mice (**Figure 3A**). Interestingly, IL-23-independent Th17 differentiation and maintenance is also known, which typically is mediated by PGE2 (**Hernandez et al., 2015**; **Paulissen et al., 2013**). Also, PGE2 synergizes with IL-23 to promote the expansion of Th17 cells (**Chizzolini et al., 2008**). While IL-23 did not show an increase, we observed that there was significantly more PGE2 in the lung homogenates of *Mtb*-infected *ifng*-/- mice (**Figure 3B**). Interestingly, both the IL-17 pathway and eicosanoid pathway were also enriched in the neutrophil's RNAseq analysis (**Figure 1G–H**). Thus, we wondered whether cells other than Th17 could be responsible for increased IL-17 production. We analyzed the IL-17-producing cells in the lungs of uninfected (**Figure 2—figure supplement 1F**) or *Mtb*-infected WT mice (**Figure 3C**). In the CD45+CD11b-Ly6G- population, which mostly represented adaptive immune cells like T cells, B cells, NK cells, etc, <100 cells were positive for IL-17 in the uninfected mice, which increased to about 4000 cells upon infection, representing about <1% of this pool of cells (**Figure 2—figure supplement 1F–G**). In contrast, in the CD45+CD11b+Ly6G+ cellular pool, 70–80% of cells were positive for IL-17, showing about <2000 cells in the uninfected mice compared to >13,000 cells in the *Mtb*-infected mice (**Figure 3D**, **Figure 2—figure supplement 1F**). Immunofluorescence analysis of lung sections from *Mtb*-infected WT mice revealed similar observations when stained for CD4 and IL-17. While we could witness CD4 and IL-17 co-expressing cells, several IL-17+ cells lacked CD4 expression (**Figure 3E**). Staining for Ly6G and IL-17 revealed intense IL-17 and Ly6G co-staining in the lung tissue sections of *Mtb*-infected WT mice, suggesting Ly6G+Gra cells as the main source of lung IL-17 (**Figure 3F**). A similar IL-17 and Ly6G co-staining was also observed in the lungs of *Mtb*-infected *ifng*-/- mice (**Figure 3G**). Interestingly, even in human TB granulomas, we observed considerable co-staining of CD66b (the human neutrophil marker) and IL-17 (**Figure 3H**). Together, our data indicate that during *Mtb* infection, an active IL-17-Ly6G+Gra axis gets established, which may contribute to TB pathology.

## Targeting IL-17 and PGE2 pathways controls bacterial burden and disease pathology in *Mtb*-infected *ifng*-/- mice

The impact of IL-17 on TB pathogenesis is not unequivocally established. For example, IL-17 is believed to exacerbate TB pathology during the chronic phase of infection, partly through excessive neutrophil recruitment (**Torrado and Cooper, 2010**). On the other hand, a more protective role of IL-17 in the WT mice was reported when infected with a hypervirulent strain of *Mtb* (**Domingo-Gonzalez et al., 2017**). In addition, PGE2 works contrastingly towards IL-17 and IFNγ production by T cells, increasing the levels of IL-17 (**Napolitani et al., 2009**). Since the results in the above sections suggested a pro-pathogenic role of IL-17 and PGE2, we decided to neutralize IL-17 and inhibit PGE2 in *Mtb*-infected *ifng*-/- mice. To inhibit PGE2 biosynthesis, we decided to block the upstream enzyme COX2, which shifts the arachidonate flux towards 5-LO, leading to increased leukotriene production (**Funk, 2001**). The *Mtb*-infected *ifng*-/- mice were administered anti-IL-17 monoclonal Ab (mAb) intra-peritoneally every 3–4 days, starting a day before the infection (**Figure 4A**). For both isotype control mAb and anti-IL17 mAb arm, celecoxib (50 mg/kg) treatment was initiated 3 weeks p.i. (**Figure 4A**). A significant decline in IL-17 levels in the lung homogenates was observed in anti-IL-17 mAb-administered mice (**Figure 4—figure supplement 1A**). Celecoxib alone had no significant impact on IL-17 levels in *ifng*-/- mice (**Figure 4—figure supplement 1A**). Combining anti-IL-17 mAb with celecoxib showed a similar decline in IL-17 levels as was noted in the anti-IL-17 mAb group alone (**Figure 4—figure**

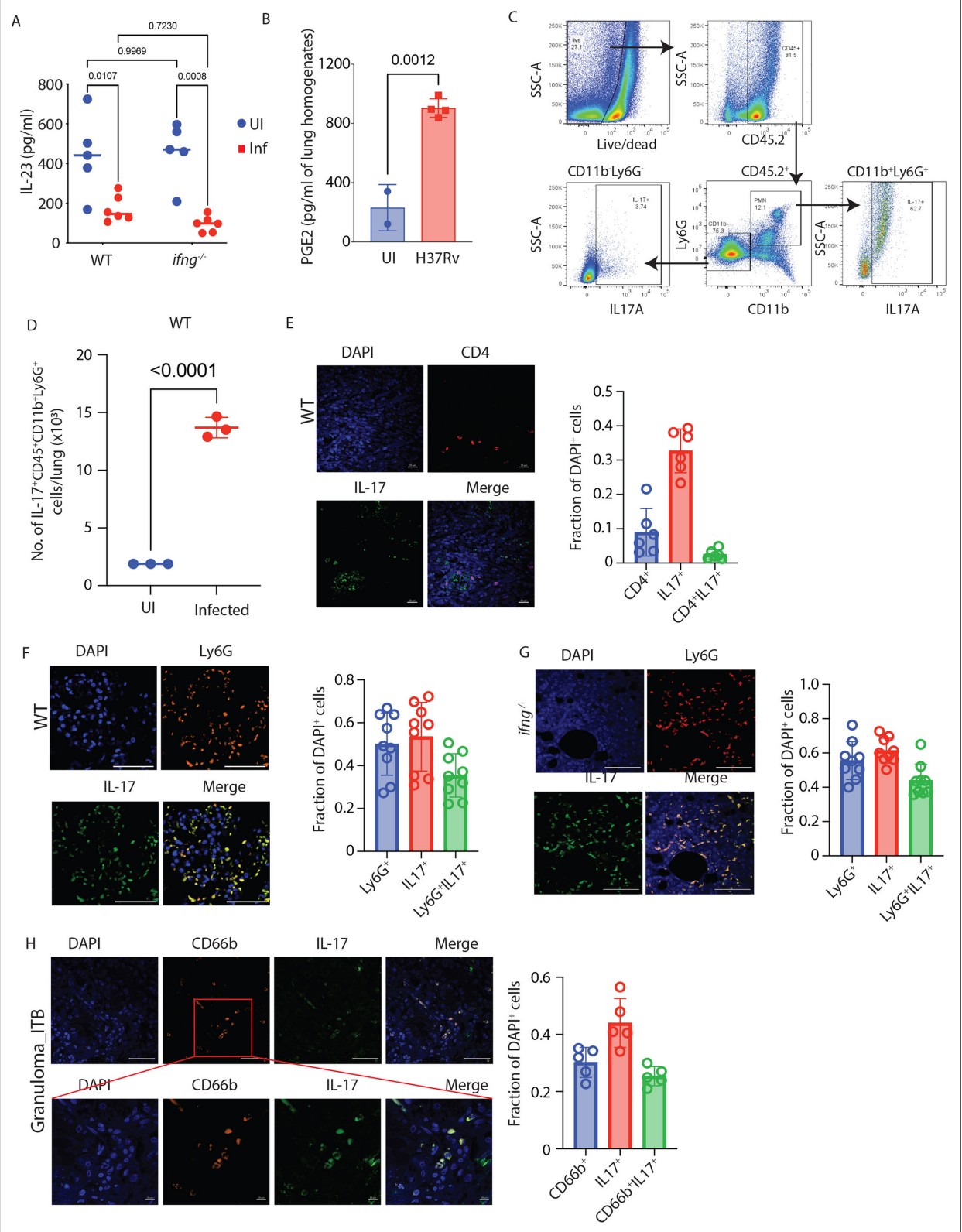

**Figure 3.** PMNs are one of the major sources of IL-17 in the lungs of *Mtb*-infected mice. (**A**) Levels of IL-23 in the lung supernatant of uninfected and *Mtb*-infected WT and *ifng*$^{-/-}$ mice. (**B**) PGE2 levels in the lungs of *Mtb*-infected *ifng*$^{-/-}$ mice compared to uninfected control. An unpaired student's t-test was applied to calculate the significance between the two groups. (**C**) Lung cells from *Mtb*-infected WT mice were stained with different antibodies and assayed through flow cytometry to elucidate the cellular source for elevated IL-17. FACS gating strategy showing PMNs (Ly6G$^+$ CD11b$^+$ cells) and

*Figure 3 continued on next page*

*Figure 3 continued*

Ly6G⁻ CD11b⁻ cells showing IL-17 positivity in the lungs of *Mtb*-infected mice. (**D**) Number of Ly6G⁺ IL-17⁺ granulocytes in the lungs of uninfected and *Mtb*-infected mice. (**E**) IFA images of the lungs of *Mtb*-infected WT mice at 6 weeks post-infection time point showing the co-localization of CD4⁺ T cells (shown in red color) with IL-17 (shown in green color). The quantification of the CD4⁺ cells, IL-17⁺ cells, and double-positive IL-17⁺ CD4⁺ cells is presented as the proportion of DAPI⁺ cells. (**F**) IFA images of the lungs of *Mtb*-infected WT mice at 6 weeks post-infection time point showing the co-localization of Ly6G⁺ Gra (shown in red color) with IL-17 (shown in green color). The quantification of the Ly6G⁺ cells, IL-17⁺ cells, and double-positive IL-17⁺ Ly6G⁺ cells is presented as the proportion of DAPI⁺ cells. (**G**) IFA images of the lungs of *Mtb*-infected *ifng⁻/⁻* mice at 6 weeks post-infection time point showing the co-localization of PMNs (Ly6G⁺ cells shown in red color) and IL-17 (shown in green color). The quantification of the Ly6G⁺ cells, IL-17⁺ cells, and double-positive IL-17⁺ Ly6G⁺ cells is presented as the proportion of DAPI⁺ cells. (**H**) Co-localization of PMNs (CD66b⁺ cells shown in red color) and IL-17 (shown in green color) in the gut granuloma section of intestinal TB (ITB) patients. The quantification of the CD66b⁺ cells, IL-17⁺ cells, and double-positive IL-17⁺ CD66b⁺ cells is presented as the proportion of DAPI⁺ cells.

supplement 1A). Neutralizing IL-17 led to a significant decline in the number of infiltrating Ly6G⁺Gra in the lungs of *Mtb*-infected *ifng⁻/⁻* mice (**Figure 4B**). Interestingly, celecoxib treatment alone also efficiently reduced the number of infiltrating Ly6G⁺Gra (**Figure 4B**). However, the combined treatment with anti-IL-17 mAb and celecoxib had no synergistic effect (**Figure 4B**). While IL-17 neutralization and celecoxib treatment alone reduced Ly6G⁺Gra recruitment in the lungs, these treatments alone could not bring down the *Mtb* burden in the lung Ly6G⁺Gra from *ifng⁻/⁻* mice (**Figure 4C**). However, their combination significantly reduced the *Mtb* burden in the lung Ly6G⁺Gra from *ifng⁻/⁻* mice (**Figure 4C**). A similar trend was also seen for the whole lung CFU, where anti-IL17 mAb or celecoxib alone could not significantly reduce the *Mtb* burden, the combined treatment showed significant decline in *Mtb* population (**Figure 4D**). Unlike in the lungs, or sorted Ly6G⁺Gra from the lungs, IL-17 neutralization alone was sufficient to reduce bacterial burden in the spleen (**Figure 4E**), indicating Ly6G⁺Gra's potential role in *Mtb* dissemination. We also noticed a significant rescue in the survival of *ifng⁻/⁻* mice by combined IL-17 neutralization and celecoxib treatment (**Figure 4F**). This aligned with reduced lung pathology in anti-IL-17 mAb and celecoxib-treated *Mtb*-infected *ifng⁻/⁻* mice (**Figure 4G**). Further analysis of the Ly6G⁺ cell population revealed a significant population of CD11b^{mid}Ly6G^{hi} cells in the *Mtb*-infected *ifng⁻/⁻* mice, which typically reflects an immature neutrophil population (*Scapini et al., 2016*; **Figure 4H**). A reduction in this population coincided with the protective effects of anti-IL-17 mAb and celecoxib treatment. However, such distinction was not very apparent in the WT mice. Taken together, our data suggest the protective effects of combined treatment with anti-IL-17 mAb and celecoxib in TB by reducing the Ly6G⁺Gra population.

## Pathological role of Ly6G⁺Gra-derived IL-17 in the lungs of *Mtb*-infected *ifng⁻/⁻* animals

We next investigated the effect of IL-17 neutralization or celecoxib treatment on lung IL-17-producing cells in *Mtb*-infected *ifng⁻/⁻* mice. We first assessed the number of IL-17-positive T helper cells (Th17) in the single-cell preparations from the lungs. While there was a trend of a decline in the number of Th17 cells upon either IL-17 neutralization alone or in combination to COX2 inhibition in the lungs of *Mtb*-infected *ifng⁻/⁻* mice, those changes were not found significant (**Figure 4—figure supplement 1B**). In the lung sections stained for IL-17 along with Ly6G, IL-17 neutralization, either alone or combined with COX2 inhibition, led to a consistent decline in IL-17⁺ Ly6G⁺ cells (**Figure 4H**). The results from IL-17 ELISA in the lung homogenates, Th17 cell count in the lungs, and IL-17-Ly6G co-staining together suggest that while IL-17 contributes to high Ly6G⁺Gra infiltration, it is the Ly6G⁺Gra that produces IL-17 locally, thereby establishing a vicious loop.

## Pathological role of Ly6G⁺Gra-IL-17 axis in WT animals

The WT animals also show an increase in IL-17 levels upon *Mtb* infection, albeit to a lower magnitude than the *ifng⁻/⁻* mice (**Figure 5A**). Moreover, as noted in **Figure 3F**, the WT mice also showed substantial IL-17-Ly6G co-staining. Hence, we asked whether a similar IL-17-Ly6G⁺Gra axis could also be operating in the *Mtb*-infected WT animals and contribute to sustained Ly6G⁺Gra-resident pool of *Mtb* in the lungs. To test this hypothesis, we decided to inhibit IL-17 in *Mtb*-infected WT mice. Instead of using anti-IL-17, we decided to use SR2211 (**Figure 5B**), a pharmacological inhibitor of RORγt (inverse agonist), the main transcription factor for IL-17 expression (*Berry et al., 2010*; *Tan et al., 2013*), which may provide improved clinical benefits compared to anti-IL-17 mAb (*Lamb et al., 2021*).

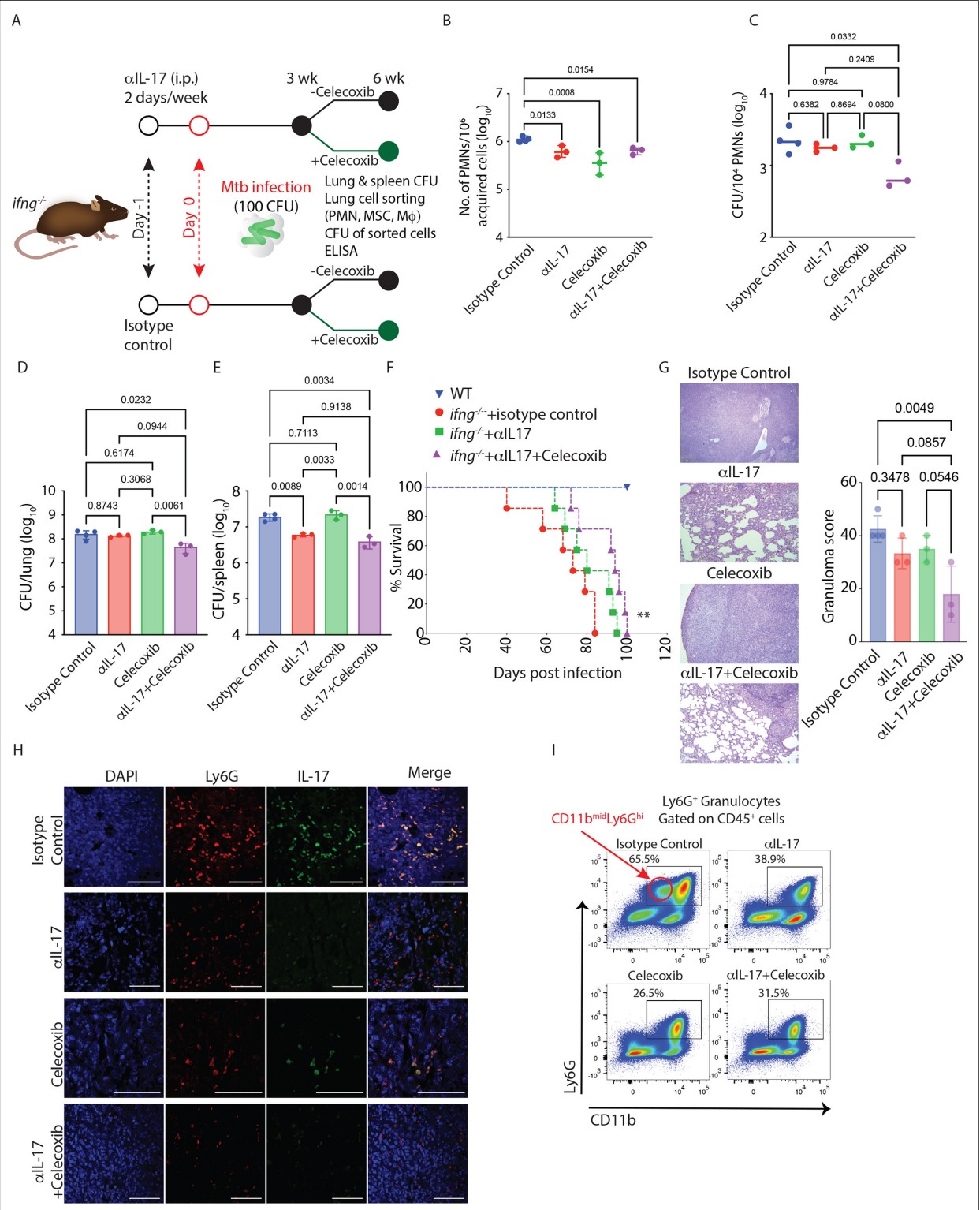

**Figure 4.** Celecoxib treatment with IL-17 neutralization reverses IL-17-dependent neutrophilia and controls TB disease in *ifng⁻/⁻* mice. (**A**) Study plan. IFNγ *⁻/⁻*C57BL/6 mice infected with H37Rv at a dose of 100 CFU via the aerosol route. One arm of the *Mtb*-infected mice received an αIL-17 neutralizing antibody (100 µg/mouse), and the other arm received an isotype control antibody (100 µg/mouse) intraperitoneally twice in a week, starting a day before *Mtb* infection. After 3 weeks of infection, both groups of mice were further divided into two groups, one that received celecoxib (50 mg/kg, orally) for 3 weeks and the other that received vehicle control (DMSO here). Mice from all four groups of mice were euthanized at 6 weeks post-*Mtb* infection, and their lungs and spleen were harvested, and serum was stored for ELISA. (**B**) PMN count, normalized to 1 million total acquired cells, in the lungs

*Figure 4 continued on next page*

*Figure 4 continued*

of infected mice in all four treatment groups at 6 weeks post-infection. (**C**) CFU (normalized to per $10^4$ sorted cells) inside PMNs sorted out from the lungs of infected mice at 6 weeks post-infection sacrifice. Bacterial burden in lung (**D**) and spleen (**E**) of *Mtb*-infected mice from described treatment groups at 6 weeks post-infection time point sacrifice. The statistical significance of the treatment groups was assessed by applying a one-way ANOVA test. (n=3–4 mice/group). This experiment was performed only once as a single experimental replicate. (**F**) Survival curve of the group of *Mtb*-infected *ifng$^{-/-}$* mice that received αIL-17 neutralizing antibody, celecoxib alone or together, along with the untreated control group and infected WT mice group. Both mice strains were infected with 100 CFU of H37Rv through an aerosol route. One arm of the *Mtb*-infected mice received an αIL-17 neutralizing antibody, and the other arm received an isotype control antibody twice a week, starting a day before the *Mtb* infection. After 3 weeks of infection, both groups of mice were further divided into two groups, one that received celecoxib for 3 weeks and the other that received vehicle control (DMSO). Mice were incubated till their natural death. The time of death of each mouse was noted and plotted for survival analysis (n=7 mice/group). This experiment was performed only once as a single experimental replicate. (**G**) H&E-stained lung sections and granuloma scoring showing lung pathology from all four treatment groups. (**H**) Lung IFA images from the above-stated treatment groups of mice show the colocalization (yellow) of PMNs (Ly6G$^+$ cells shown in red color) and IL-17 (shown in green color). (**I**) Representative FACS dot plot showing the percentage of PMNs in the lungs of *Mtb*-infected mice in all four above-stated treatment groups (population is gated on CD45$^+$ cells). This also shows the presence of a distinct CD11b$^{mid}$ Ly6G$^{hi}$ population in the isotype control group.

The online version of this article includes the following figure supplement(s) for figure 4:

**Figure supplement 1.** IL-17 neutralization synergizes with COX-2 inhibition and controls TB disease in *ifng$^{-/-}$* mice.

We also included the celecoxib group since *Mtb*-infected WT mice also show very high PGE2 levels in the lung homogenates (*Figure 5C*). Treatment with SR2211 led to a significant decline in IL-17 levels in the lung homogenates of *Mtb*-infected WT mice (*Figure 5D*). Interestingly, unlike the *Mtb*-infected *ifng$^{-/-}$* mice, celecoxib alone was also able to reduce the level of IL-17 in the lung homogenates of *Mtb*-infected WT mice, and so was the combination of both SR2211 and celecoxib (*Figure 5D*). Treatment with SR2211 showed a downward trend in Ly6G$^+$Gra count in the lungs; however, the differences were not significant (*Figure 5E*, *Figure 5—figure supplement 1A*). At the whole tissue level, SR2211 alone did not reduce the *Mtb* burden in the lungs significantly (*Figure 5F*). However, it was highly efficient in controlling dissemination to the spleen (*Figure 5G*). Similarly, celecoxib treatment, either alone or in combination with SR2211, significantly reduced bacterial CFUs in the lung and spleen; however, the effects were more pronounced in the spleen (*Figure 5F–G*). Furthermore, SR2211 and celecoxib, either alone or in combination, led to a significant decline in the Ly6G$^+$Gra-resident *Mtb* population (*Figure 5H*). The histopathology analysis of the lungs reflected the protective effects of SR2211 and celecoxib (*Figure 5—figure supplement 1B–C*). Moreover, SR2211 treatment also led to a loss of IL-17-Ly6G co-staining, indicating the role of Ly6G$^+$Gra-derived IL-17 in disease pathology in the WT mice (*Figure 5I*). These results in the WT mice further emphasize the establishment of the IL-17-Ly6G$^+$Gra axis during *Mtb* infection and demonstrate its role in TB pathology and disease.

## Modulating IL-17 via COX2 inhibition or by RORγt inhibition enhances bacterial control in BCG-vaccinated animals by lowering Ly6G$^+$Gra-resident *Mtb*

Our above data in WT and *ifng$^{-/-}$* mice indicate IL-17-producing Ly6G$^+$Gra as one of the drivers of TB pathology. We next evaluated whether IL-17 inhibition could also help enhance the efficacy of BCG-mediated control of TB in WT mice. For IL-17 inhibition, we used celecoxib treatment since it effectively reduced the IL-17 levels in the lungs of WT mice (*Figure 5D*). BCG-vaccinated and PBS control WT mice were challenged with H37Rv (100–200 CFU, *Figure 6A*) at 8 weeks post-immunization. Celecoxib treatment was started four weeks after the infection, and we collected samples at 8 and 12 weeks p.i. Celecoxib treatment significantly enhanced the efficacy of BCG against *Mtb* challenge, leading to a reduced bacterial burden in the lung of BCG +Celecoxib mice compared to BCG-alone mice at 12 weeks p.i. (*Figure 6B*, left panel). BCG +Celecoxib, however, did not show a significant advantage over the BCG-alone group for spleen *Mtb* load (*Figure 6B*, right panel). Importantly, the BCG +Celecoxib group also showed significantly reduced *Mtb* burden in lung Ly6G$^+$Gra or PMN cells compared to the BCG-alone group (*Figure 6C*). Celecoxib treatment led to a decline of ~45% in the *Mtb* burden in lung Ly6G$^+$Gra (*Figure 6C*), which also corresponded to better histopathology (*Figure 6D–E*). Similar observations were also made when IL-17 was directly inhibited using SR2211 (*Figure 6F–H*). Together, we conclude that the IL-17-Ly6G$^+$Gra axis contributes to TB pathology and

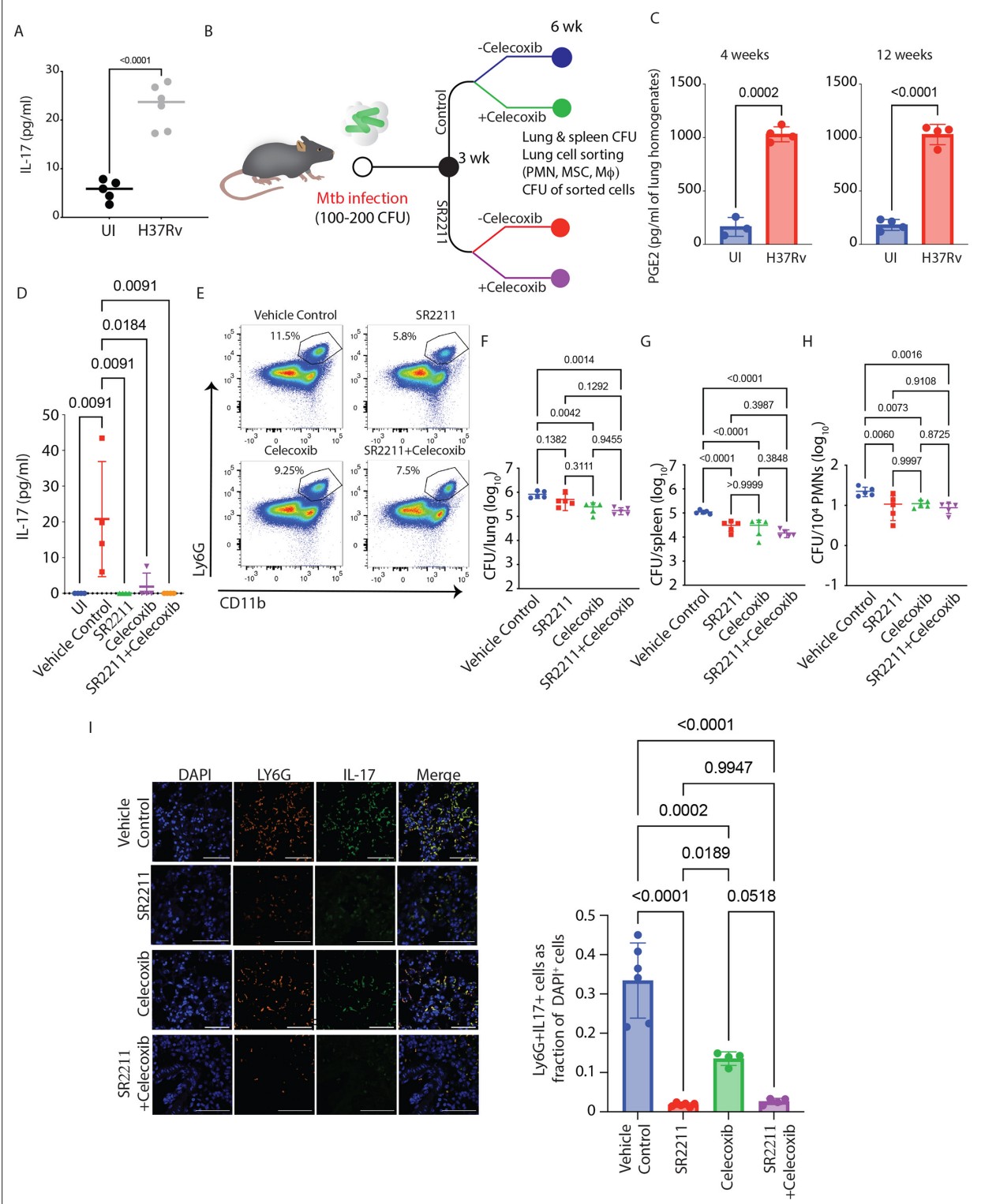

**Figure 5.** RORγt inhibition reduces mycobacterial burden and neutrophil infiltration in the lungs of *Mtb*-infected mice. (**A**) IL-17 levels in the lung homogenates of uninfected and *Mtb*-infected WT mice. An unpaired student's t-test was applied to calculate significance between the two groups. Sample size (n)=5–6 mice/group (**B**) Study plan. C57BL/6 mice were infected with H37Rv at a 100–200 CFU dose via the aerosol route. At 3 weeks post-infection, treatment with celecoxib and SR 2211 (inverse agonist for RORγt) was started for a further 3 weeks. Celecoxib (50 mg/kg) was administered orally once a day, and SR 2211 (20mg/kg) was administered intraperitoneally thrice in a week. At 6 weeks post-infection, mice from the treatment and untreated control groups were sacrificed, lung and spleen were harvested for CFU, histopathology, and FACS analysis, and lung homogenates were

*Figure 5 continued on next page*

*Figure 5 continued*

stored for ELISA. (**C**) PGE2 levels in the lung homogenates of uninfected and *Mtb*-infected WT mice at 4 and 12 weeks post-infection time point. Unpaired student's t-test was applied to calculate significance between the two groups at each time point. Sample size (n)=3–4 mice/group. (**D**) Levels of IL-17 in the lung homogenates of all four-treatment groups of mice. The statistical significance of the treatment groups was assessed by applying a one-way ANOVA test, n=3–4 mice/group. (**E**) Representative FACS dot plot showing the percentage of PMNs in the lungs of *Mtb*-infected mice in all four above-stated treatment groups (population is gated on $CD45^+$ cells). Mycobacterial burden in the lung (**F**) and spleen (**G**), in the above-stated treatment groups of mice. (**H**) Lung PMNs *Mtb* burden normalized to per $10^4$ sorted cells in all four treatment groups of mice. The statistical significance of the treatment groups was assessed by applying a one-way ANOVA test, n=5 mice/group. (**I**) Lung IFA images from the above-stated treated group of mice show the colocalization of PMNs ($Ly6G^+$ cells shown in red color) and IL-17 (shown in green color). The $IL-17^+$ $Ly6G^+$ (double-positive) cells were quantified and represented as a percentage of $DAPI^+$ cells in all four treatment groups. This experiment was performed only once as a single experimental replicate.

The online version of this article includes the following figure supplement(s) for figure 5:

**Figure supplement 1.** IL-17 production inhibition is key to limiting TB severity in WT mice.

interferes with the efficacy of the BCG vaccine and, therefore, could be targeted to achieve better control of TB.

## The IL-17-Granulocyte axis in human TB subjects

The results so far establish the role of the IL-17- $Ly6G^+$Gra axis in TB pathology in mice. To verify the existence of a similar axis in human TB, we mined data from a prospective cohort study of pulmonary tuberculosis (PTB). This cohort comprised 201 subjects of PTB, who were treated with anti-TB treatment (ATT) regime for 6–8 months and followed up for an additional 1 year to track any recurrence (*Figure 7A*; *Kumar et al., 2021*; *Kumar et al., 2020*). Among those, 133 subjects showed no recurrence of TB at one year of follow-up and were therefore considered cured of PTB, whereas 68 subjects failed the treatment and showed manifestation of TB recurrence and death (*Figure 7A*). We compared blood IL-17 and neutrophil levels at the time of recruitment (under treatment-naïve conditions) among these subjects of the cohort. The failed treatment group had significantly higher serum IL-17 levels and absolute blood neutrophil count at the time of recruitment compared with those who were successfully cured from the disease and showed no recurrence (*Figure 7B–C*). Moreover, the treatment failure group also had a significantly higher neutrophil-to-lymphocyte ratio (NLR) at the time of recruitment (*Figure 7D*). Higher NLR has been demonstrated to be a risk of TB and associated with poor treatment outcomes (*Han et al., 2018*; *Miyahara et al., 2019*; *Nakao et al., 2019*; *Suryana et al., 2022*). Taken together, our data suggests that the IL-17-neutrophil axis influences TB pathogenesis both in mice and humans, thereby providing an opportunity to develop a protective strategy by targeting this axis (*Figure 7E*).

## Discussion

The immune response elicited by *Mtb* infection triggers the recruitment of diverse innate and adaptive immune cells to the site of infection, leading to the formation of granulomas (*Ramakrishnan, 2012*). In the granulomas, *Mtb* coopts these immune cells as its niche and evades adaptive immune response during the chronic phase (*Chandra et al., 2022*; *Lee et al., 2020*; *Lovewell et al., 2021*). Different monocyte and macrophage subpopulations in the lungs are reported as major immune cells harboring the intracellular *Mtb* in the lungs of infected mice, which may also impact *Mtb* growth and survival differentially (*Lee et al., 2020*; *Huang et al., 2018*). Our observation that $Ly6G^+$Gra, which typically represents the neutrophil or PMN population, is the major *Mtb*-infected pool in the lungs brings a new dimension to the cellular niches of *Mtb*. Previously, it was reported that the highest number of bacilli is present in the neutrophils early during the infection (*Huang et al., 2018*). Yet another study showed $Ly6G^+$ granulocytes as a permissive niche to *Mtb* (*Lovewell et al., 2021*; *Mishra et al., 2017*). However, so far, no literature suggests that the neutrophil-resident population could be responsible for lower vaccine efficacy. Thus, the inability of BCG vaccination to clear the intracellular *Mtb* burden from $Ly6G^+$Gra has serious implications. Both pro- and antibacterial roles are attributed to granulocytes, specifically neutrophils, largely due to heterogeneity in their population (*Leliefeld et al., 2018*; *Tsuda et al., 2004*). While certain innate immune responses of BCG-vaccinated animals are attributed to trained immunity (*Kaufmann et al., 2018*), it is also reported that BCG vaccination results in

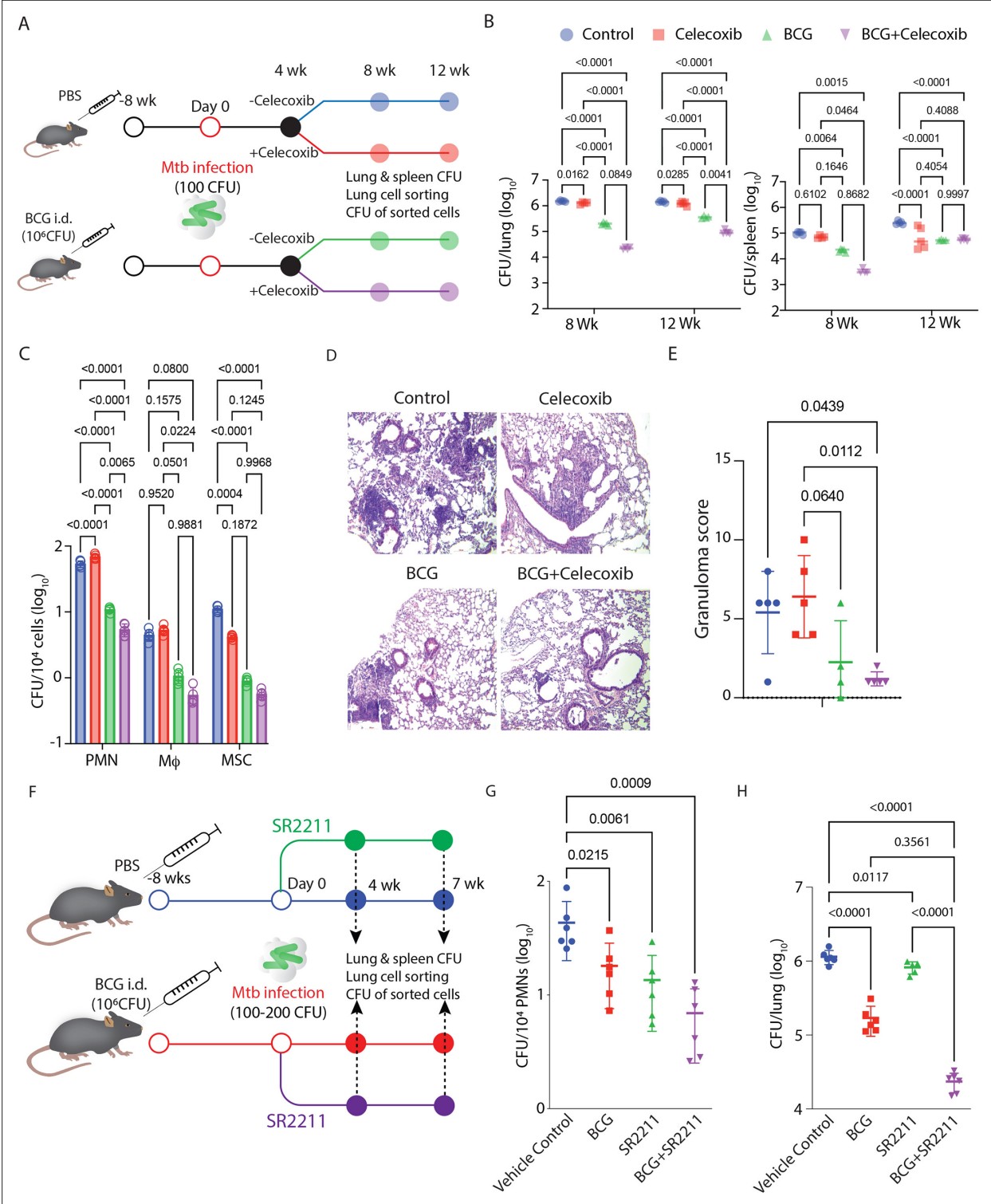

**Figure 6.** Targeting IL-17 via COX2 inhibition and RORγt inhibition augments BCG efficacy by lowering Ly6G+Gra-resident *Mtb*. (**A**) Study plan. C57BL/6 mice were vaccinated intradermally with $10^6$ BCG bacilli. After 8 weeks, vaccinated and unvaccinated mice were subjected to aerosol infection with H37Rv (100 CFU). 4 weeks post-*Mtb* infection, mice received celecoxib (50 mg/kg, orally) treatment for 4 and 8 weeks in with and without BCG vaccination group, keeping the untreated arm as control. At 4, 8, and 12 weeks post-infection, mice were euthanized, and lung and spleen were harvested. (**B**) Enumeration of lung and spleen mycobacterial burden in all four treatment groups (control, BCG-vaccinated, celecoxib-treated, and BCG-vaccinated celecoxib-treated) at 8 and 12 weeks post-infection. (**C**) *Mtb* burden in individual sorted cells (PMNs, macrophages, and MSCs), plotted as CFU/$10^4$ sorted cells at 12 weeks post-infection time point, from the lungs of all four treatment groups of mice. A two-way ANOVA statistical test

*Figure 6 continued on next page*

*Figure 6 continued*

was applied to compare all four treatment groups at different time points. Sample size (n)=5 mice/group in all treatment groups. (**D**) H&E-stained lung sections of control, BCG-vaccinated, celecoxib-treated, and BCG-vaccinated celecoxib-treated group at ×10 magnification. (**E**) Granuloma scoring of the same treatment groups depicting histopathology. This experiment was independently conducted in three separate experimental replicates (**F**) Study plan. C57BL/6 mice were vaccinated intradermally with $10^6$ BCG bacilli. After 8 weeks, vaccinated and unvaccinated mice were subjected to aerosol infection with H37Rv (100 CFU). At 4 weeks post-Mtb infection, mice were administered SR2211 (20 mg/kg) intraperitoneally three times per week for 3 weeks, in both BCG-vaccinated and unvaccinated groups, with untreated mice serving as vehicle control. At 7 weeks post-infection, mice were euthanized, and lung and spleen were harvested. (**G**) The Mtb burden in sorted PMNs was assessed at 7 weeks post-infection and expressed as CFU per $10^4$ sorted cells from the lungs of all four treatment groups (Control, BCG-vaccinated, SR2211-treated, and BCG-vaccinated+SR2211-treated). (**H**) Lung mycobacterial burden was also quantified across all four groups at the same time point. A two-way ANOVA was performed to compare the groups. Sample size (n)=6 mice/group in all treatment groups. This experiment was performed only once as a single experimental replicate.

neutrophil reprogramming, leading to an enhanced antibacterial state (*Moorlag et al., 2020*). We did observe BCG-mediated decline in the Ly6G+Gra-resident *Mtb* population, suggesting an antibacterial activation state. However, they continued to represent the largest, if not the only, pool of intracellular bacilli in those animals. Our observations were in line with some human studies, where in TB patients, neutrophils were the most predominant infected phagocytes in the sputum samples, BAL fluid, and lung cavities (*Eum et al., 2010*).

Granulocyte recruitment to the site of inflammation is driven by chemotactic stimuli like CXCL1, CXCL2, CCL3, etc that trigger neutrophil chemotaxis under resting or activated states via the chemotactic receptor CXCR2 (*Kolaczkowska and Kubes, 2013*) and the eicosanoids downstream to COX2 (*Funk, 2001*; *Lemos et al., 2009*). In the *ifng*−/− mice, the neutrophil chemotactic signals were upregulated to a varying degree upon *Mtb* infection. In addition, IL-17 is also implicated in neutrophil recruitment across different settings (*Flannigan et al., 2017*; *Gelderblom et al., 2012*). In TB, IL-17 from CD4+ T cells is believed to trigger neutrophil infiltration (*Nandi and Behar, 2011*). Interestingly, IFNγ can block IL-17 production and, therefore, serve as the mechanism for high neutrophilia in *ifng*−/− mice upon *Mtb* infection (*Nandi and Behar, 2011*). Interestingly, IFNγ can also trigger nonhematopoietic cells in the lung epithelial and endothelial compartments to promote IDO expression, which restrains IL17 production and neutrophil-driven inflammation, thus limiting *Mtb* burden (*Desvignes and Ernst, 2009*). We also observed high Ly6G+Gra infiltration and lung IL-17 levels in *Mtb*-infected *ifng*−/− mice. However, the source of lung IL-17 appeared to be distinct from the typical Th17 cells, consistent with a lack of any increase in IL-23 levels, important for the expansion of Th17 cells (*Haines et al., 2013*; *Langrish et al., 2005*; *Teng et al., 2015*). Instead, our results show that Ly6G+Gra are a major source of IL-17 in *ifng*−/− mice as well as in WT mice. Since Ly6G+Gra are also known to express IL-17 receptors (*Tan et al., 2013*; *Taylor et al., 2014*), these results raise the possibility that IL-17 and Ly6G+Gra could establish a self-amplification loop, leading to severe TB pathology.

Including a COX2 inhibitor in the experiments with WT mice and *ifng*−/− mice was very insightful. The crosstalk between prostaglandins and IL-17 is reported as they help Th17 cell differentiation (*Napolitani et al., 2009*). The COX2 enzyme competes with the 5-lipoxygenase (5-LO) enzyme for the substrate arachidonic acid to promote prostaglandin and thromboxane biosynthesis, and inhibiting COX2 redirects arachidonate towards the leukotriene pathway (*Funk, 2001*). To note, increased production of leukotriene is also associated with TB protection (*Pavan Kumar et al., 2019*; *Peres et al., 2007*; *Tobin et al., 2010*). While COX2 inhibition could not significantly impact IL-17 levels in the *ifng*−/− mice, it was very effective in controlling IL-17 levels in the WT mice. Combining IL-17 neutralization with COX2 inhibition showed better *Mtb* control in the *ifng*−/− mice. Inhibiting RORγt with SR2211 provided insights into the crosstalk between IL-17 and COX2 in the WT mice. RORγt is a classic nuclear receptor implicated in the Th17 cell differentiation (*Ivanov et al., 2006*). In the WT mice, while RORγt inhibition or COX2 inhibition alone significantly lowered *Mtb* burden in the lung Ly6G+Gra, no synergistic or additive effect on *Mtb* burden was seen upon combined treatment with SR2211 and celecoxib, suggesting that COX2 and IL-17 act in a common pathway. To note, only in the WT and not in the *ifng*−/− mice, COX2 inhibition completely abrogated lung IL-17 levels, suggesting COX2 largely worked through IL-17 regulation in the *Mtb*-infected WT mice.

Our observation of Ly6G+Gra as a major source of IL-17 during *Mtb* infection in both WT and *ifng*−/− mice is unprecedented. Granulocytes, specifically neutrophils, are known to express COX2 under inflammatory conditions and secrete PGE2 (*Maloney et al., 1998*). Also, PGE2 can regulate neutrophil migration, specifically when induced by IL-23/IL-17 (*Lemos et al., 2009*). However, we suspect

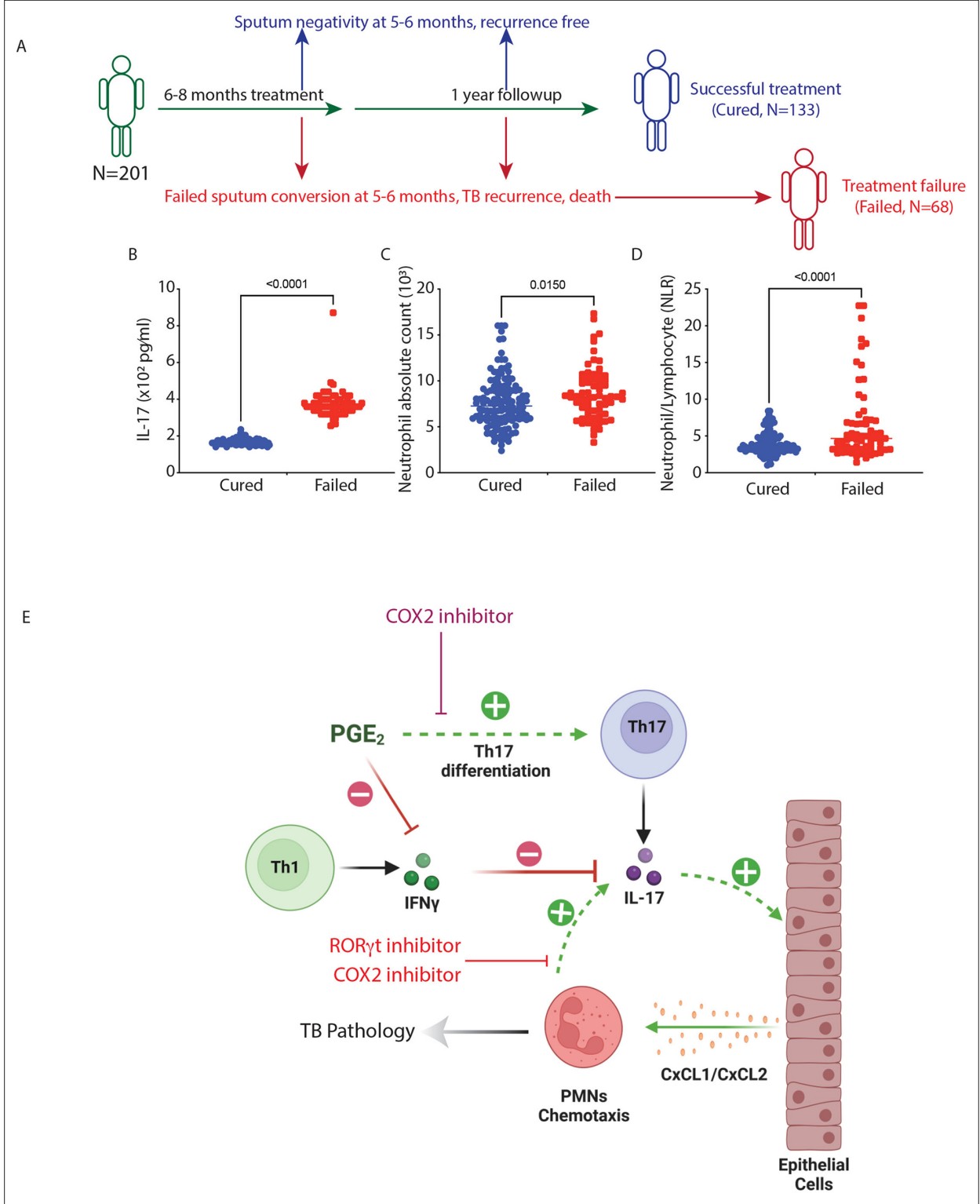

**Figure 7.** High IL-17 precipitates adverse treatment outcomes in pulmonary TB patients. (**A**) Pulmonary TB (PTB) patients-cohort study design: It includes 133 subjects that were cured of TB after the full standard ATT regime and 68 subjects that have failed the standard ATT regime (independent of drug resistance) or have experienced recurrence of symptoms. At the time of recruitment in the cohort study, the baseline blood levels of IL-17, neutrophils, and total lymphocytes in both the treatment groups were measured. (**B**) Blood IL-17 levels, (**C**) blood neutrophil levels, and (**D**) blood neutrophil/total lymphocyte ratio in treatment failure and treatment success cases of pulmonary TB patients. Unpaired student's t-test was applied to calculate significance. (**E**) Schematic showing the therapeutic potential of COX-2 and RORγt inhibitor in TB by targeting IL-17-PMN axis.

PGE2 also somehow curtails the antibacterial responses of neutrophils since we observed conditions where, without a drop in neutrophil count, the intra-Ly6G+Gra bacillary load was reduced significantly upon COX2 inhibition.

Neutrophils express RORγt and IL-17 and can be activated through autocrine IL-17A-IL-17RC interactions during fungal infections (*Taylor et al., 2014*). However, unlike our results, the IL-17-mediated autocrine activation of neutrophils was shown to elicit strong anti-microbial functions in the neutrophils (*Taylor et al., 2014*). It is possible that microbial species and infection sites could drive these responses differentially. Thus, our results suggest that IL-17-Ly6G+Gra crosstalk restrains the protective host responses in normal and highly susceptible models of TB.

As reported earlier, the Ly6G+Gra-resident pool of *Mtb* showed resilience to BCG vaccination. However, COX2 inhibition in the BCG-vaccinated WT mice significantly reduced the bacterial burden and disease pathology. Since SR2211 also worked in a similar way as COX2 in enhancing BCG vaccine efficacy, and under both conditions, IL-17 levels were dramatically reduced in the lung homogenates, the IL-17-mediated mechanism in vaccine efficacy appears plausible. Alternatively, it is reported that BCG-dependent COX-2/PGE2 pathway activation blunts macrophage, DC, and PMN responses (*Ballinger et al., 2006*; *Serezani et al., 2007*; *Sheppe and Edelmann, 2021*). This has led to explorations on a combination of BCG with selective COX2 inhibitors to ameliorate the efficacy of BCG (EP2763666B1; *Ibrahim et al., 2021*). Curiously, selective COX-2 inhibition by celecoxib had no impact on neutrophil numbers in the lungs; it significantly reduced the *Mtb* load in the neutrophils in BCG-vaccinated animals. Thus, our results show that the COX2 pathway contributes to lowering the efficacy of BCG-mediated protection by preventing the antibacterial functions of neutrophils. The COX-2-mediated negative regulation of neutrophil antibacterial response is also shown in the case of *Pseudomonas aeruginosa* infection (*Sadikot et al., 2007*).

The results on the human cohort re-emphasize the primacy of neutrophils in determining the disease outcome in humans. We found that high neutrophils and IL-17 can predict treatment failure. This underscores a similar neutrophil-IL-17 axis operational in humans underlying disease severity, as observed in the mouse model in this study. Interestingly, a neutrophil-driven transcriptional signature was earlier reported to be associated with active TB (*Berry et al., 2010*). It is noteworthy that high neutrophilia alone has been shown to predict death in TB patients (*Lowe et al., 2013*).

The results from mice lacking IFNγ have immediate clinical application. A relatively small fraction but a sizable number of TB cases belong to a category known as Mendelian Susceptibility to Tubercular Diseases or MSMD, which typically is caused by a mutation in the IFNγ/IL12p70 pathway (*Filipe-Santos et al., 2006*; *Kerner et al., 2020*; *Levin et al., 1995*). These patients develop severe diseases even upon exposure to BCG and respond poorly to anti-TB treatment (*Casanova et al., 1995*; *Mahdaviani et al., 2022*). We believe the MSMD phenotype could be a manifestation of the IL-17-neutrophil axis observed in *ifng*-/- animals, which could be exploited for better treatment outcomes in these patients.

One clear limitation of the study is the lack of a detailed characterization of the diverse granulocyte populations in the lungs. For example, experiments in the *ifng*-/- suggested a potential pathological role of CD11bmidLy6Ghi granulocytes. The heterogeneity among granulocytes is well reported, and emerging evidence suggests neutrophils are detrimental to the host, while eosinophils are protective to the host against TB (*Mayer-Barber, 2023*). We believe further characterization of this diverse population could provide greater insights into the protective and pathological roles played by granulocytes. Another shortcoming is the lack of direct evidence of IL-17 secretion by PMNs in the lung granulomas, which would potentially require PMN-specific genetic deletion of IL-17.

Finally, there is a lack of consensus on the protective or pathological role of IL-17 in TB. The protective effect of IL-17 has been shown in the mouse model of TB in the context of vaccination (*Choi et al., 2020*; *Desel et al., 2011*; *Khader et al., 2007*). Ironically, in the majority of such studies, the IL-17 response was measured through ex vivo stimulation of isolated T cells. In contrast, in this study, we measured IL-17 levels in the lung homogenates, which could more accurately capture the function of this cytokine in the TB granulomas in situ.

To conclude, through this study, we experimentally establish Ly6G+Gra as a niche for *Mtb* under normal and severe disease conditions, largely facilitated by IL-17 and the COX2 pathways. Targeted inhibition of IL-17 or COX2 can be exploited to enhance the efficacy of BCG or other vaccine

candidates and as adjunct therapeutic strategies for the normal population and among those with genetic vulnerability towards severe TB.

## Materials and methods

### Ethical clearance

ITB biopsy samples and control samples were obtained from the Department of Pathology, AIIMS, New Delhi. These biopsy samples were taken from the patients for diagnostic purposes after taking their written informed consent. The use of the archived leftover biopsy samples was approved by the institute's EC of AIIMS (Ref no. IEC-304/02-06-2017), and ICGEB (Ref no. ICGEB/IEC/2017/06-verII and ICGEB/IEC/2016/03) and the same were accessed according to the institutional guidelines. Animal experiments were approved by Institutional Animal Ethics Committee, ICGEB (ICGEB/IAEC/280718 / CI-14), (ICGEB/IAEC/07032020 /CI-18), and (ICGEB/IAEC/18092021 /CI-16). The study on human PTB subjects was approved by the Ethics Committees of the Prof. M. Viswanathan Diabetes Research Center (ECR/51/INST/TN/ 2013/MVDRC/01) and NIRT (NIRT-INo:2014004). Informed written consent was obtained from all participants. All the methods were performed in accordance with the relevant institutional ethical committee guidelines.

### Reagents and antibodies

Tween-20 (P1379), Vectashield (F6182), Triton-X-100 (9036-19-5), DNase 1 (D5025), Collagenase D (11088858001), Propidium Iodide (P4170), HBSS (H6648), Trizol (T9424), Hematoxylin (51275), Eosin (318906), Formalin (HT501128), Celecoxib (SML3031-50MG), DAPI (28718-90-3), Xylene (CAS-1330-20-7), Absolute ethanol (CAS- 64-17-5), Sodium Citrate (CAS-–6132-04-3) were obtained from Sigma-Aldrich. SR 2211, Selective RORγt inverse agonist (4869) was procured from TOCRIS. 7H9 (271310), 7H11 (212203), ADC (211887), OADC (211886) FACS Flow Sheath Fluid (342003), Accudrop Fluorescent Beads (345249), CS&T Research Beads Kit (655050), all FACS antibodies used for acquisition and sorting (details are provided in the separate table) were procured from Becton Dickinson Private Limited. Fixable Viability Dye eFluor 780 used for live/dead staining was obtained from eBioscience. InVivoMAb anti-mouse IL-17A (BE0173), InVivoPure pH 7.0 Dilution Buffer (IP0070), InVivoMAb mouse IgG1 isotype control, unknown specificity (BE0083), InVivoPure pH 6.5 Dilution Buffer (IP0065) were purchased from BioXcell. Mouse IL-17A ELISA Kit (ab199081), mouse PGE2 ELISA Kit ab287802 and Cytokine & Chemokine 36-Plex Mouse ProcartaPlexTM Panel 1 A (EPX360-26092-901) for Luminex assay were obtained from Abcam and Thermo Fisher, respectively. Penta antibiotic mix (FD260-5VL) was procured from Himedia. Rabbit polyclonal *Mycobacterium tuberculosis* Ag85B (ab43019) antibody was purchased from Abcam.

### Mycobacterial cultures

Bacterial strains: Lab-adapted strain of *Mycobacterium tuberculosis*, H37Rv, seed stock was received from Colorado State University. BCG (Pasteur strain) was obtained from the National Institute of Immunology, New Delhi. For *Mtb* ex vivo and in vivo infection experiments, H37Rv and BCG mycobacterial cultures were grown in 7H9 media (BD Difco) supplemented with 10% Albumin–Dextrose–Catalase (ADC, BD, Difco) and incubated in an orbiter shaker at 100 rpm, 37 °C till the growth reached the log phase. The log phase cultures were pelleted down, and the pellet was resuspended in either saline (for aerosol infection in mice) or culture media (for ex vivo infection in cells). Single-cell suspensions of H37Rv and BCG were prepared by passing bacterial cultures seven times through 23-gauge needles, five times through 26-gauge needles, and three times through 30-gauge needles.

### Animals

WT C57BL/6 mice were procured from animal house, ICGEB, New Delhi and *IFNγ-/-* C57BL/6 mice were procured from animal house, NII, New Delhi. Both male and female, 6–8 weeks old mice were used for all the in vivo experiments. *Mtb*-infected mice were housed in our biosafety level-3 laboratory and kept pathogen-free by routine serological and histopathological examinations. Animal protocols employed in this study were approved by the Institutional Animal Ethics Committee.

### Aerosol challenge in C57BL/6 mice with H37Rv strain

All mice experiments were carried out in the Tuberculosis Aerosol Challenge Facility (TACF, ICGEB, New Delhi, India). C57BL/6 mice were housed in cages within a biosafety level 3 laminar flow enclosure.

According to the standardized protocol, mice were infected with 100–200 CFU (unless otherwise stated) with H37Rv in a Wisconsin-Madison chamber. To confirm the infection establishment and to enumerate the exact CFU delivered, two animals were selected randomly and humanely euthanized 24 hr post-aerosol challenge. The lung tissues were harvested and homogenized in 1 X PBS, followed by plating on Petri plates containing Middlebrook 7H11 agar (Difco) supplemented with 10% OADC (Becton, Dickinson) 0.5% glycerol. The colonies were counted after 21 days to enumerate the *Mtb* load.

## Drug treatment and BCG vaccination of mice

COX-2 inhibitor Celecoxib (50 mg/kg) was administered to the mice through oral gavaging once per day, according to the timeline of the experiment. RORγt inhibitor SR2211 (20 mg/kg) was administered to the mice through the intra-peritoneal route every alternate day according to the timeline of the experiment. Mice were vaccinated with BCG intradermally at a dose of 1 million bacilli per mouse.

## CFU plating post-*Mtb* infection

At various time points post-*Mtb* infection, lung and spleen tissues were harvested from the infected mice. The left half lung of the mice was homogenized and serially diluted in 1 X PBS. Different dilutions were plated in duplicates on 7H11 agar plates supplemented with 10% OADC to get *Mtb* colonies within the countable range. Similarly, spleens were also harvested, homogenized, and serially diluted in 1 X PBS, followed by plating on 7H11 agar plates. The colonies were counted after 21 days to enumerate the *Mtb* load.

## FACS sample preparation

The right half of the lung of the infected mice was used in the preparation of single-cell suspension followed by FACS sorting and subsequent plating of sorted cells (i.e. neutrophils, macrophages, and mesenchymal stem cells). For sorting, the tissue was washed with 1 X PBS and chopped into small pieces. 20 U/ml DNase 1 and 1 mg/ml collagenase D were added to the chopped tissue and incubated for 30–45 min at 37 °C. After incubation, the chopped lung tissues were passed through the nylon mesh of a 70-µm cell strainer to get a single-cell suspension. The cells were pelleted at 1200 rpm, and the cell pellet was treated with 1 X ACK RBC lysis buffer to get rid of RBCs and washed with 1 X PBS. The cells were incubated with Fc-Block antibody (CD16/32 FCR-γ antibody) on ice for 15 min to inhibit the binding of surface antibodies to Fc receptors on the immune cell surface. The cells were washed with PBS and subsequently stained with the antibody cocktail of CD45.2, CD11b, Ly6G, MHCII, CD64, MerTK, Sca-1, CD90.1, and CD73 and placed on ice for 45 min. Dead cells were stained with fixable viability dye eFluor 780 (FVD eFluor 780) or Propidium Iodide (PI) at 5 µg/ml concentration just before acquisition.

## FACS cell sorting

After staining and washing, the lung cells were again passed through a 40 µm strainer before acquisition into the FACS machine to ensure formation of single-cell suspension. The whole sample was acquired to get the maximum number of live sorted cells (neutrophils, macrophages, and mesenchymal stem cells). The sorted cells were then pelleted (at 13,000 RPM), lysed, and plated on 7H11 plates. The colonies were counted after 21 days to enumerate the *Mtb* burden within each sorted cell type.

## RNA-Seq analysis

Reads were quality filtered using Trimmomatic (v.0.39, LEADING:10 TRAILING:10 SLIDING WINDOW:4:15 MINLEN:75); (*Bolger et al., 2014*) and were assessed for quality using FastQC (v.10.1; *Andrews, 2010*) and visualized with MultiQC (v.1.9; *Ewels et al., 2016*). Salmon (v.1.8, –numGibbsSamples 30; *Patro et al., 2017*) was used to quantify quality filtered reads against mouse genome (GRCm39, primary assembly) with Gencode annotation (v.M30; *Frankish et al., 2019*). Profiles were imported using tximeta (v.1.10; *Love et al., 2020*) and SummarizedExperiment (v.1.22; *Morgan et al., 2018*) which were then summarized to gene-level expression. The following biotypes were filtered out: pseudogenes, TEC, snRNA, lncRNA, miRNA, snoRNA, misc_RNA, rRNA, ribozyme, scaRNA, sRNA, scRNA, Mt_tRNA, Mt_rRNA. The dataset was further filtered to remove features not

expressed across the whole dataset. The dataset was normalized to scaled TPM values. Downstream analyses are described in the statistical analyses section.

## Multiplex microbead-based immunoassay (Luminex)

Mouse lung homogenates were collected for Luminex analysis using the Cytokine & Chemokine 36-Plex Mouse ProcartaPlexTM Panel 1 A (Thermo Fisher, # EPX360-26092-901). Samples were incubated with fluorescent-coded magnetic beads pre-coated with respective antibodies in a black 96-well clear-bottom plate overnight at 4 °C. After incubation, plates were washed five times with wash buffer (PBS with 1% BSA (Capricorn Scientific) and 0.05% Tween-20 (Promega)). Sample antibody-bead complexes were incubated with biotinylated detection antibodies for 1 hr and washed five times. Subsequently, Streptavidin-PE was added and incubated for another 30 min. Plates were washed five times before sample-antibody-bead complexes were resuspended in sheath fluid for acquisition on the FLEXMAP 3D (Luminex) using xPONENT 4.0 (Luminex) software. Data analysis was done using Bio-Plex Mana-gerTM 6.1.1 (Bio-Rad). Standard curves were generated with a 5-PL (5-parameter logistic) algorithm, reporting values for both mean fluorescence intensity (MFI) and concentration data.

## Tissue immunofluorescence assay (IFA)

Five-micron thick sections of paraffin-embedded tissue sections were taken in poly-L-lysine-coated slide. Deparaffinization was performed by heating the slides at 50 °C for 20 s till the wax melts, followed by the subsequent steps, 100% xylene for 10 min, xylene and absolute ethanol for 10 min, 100% ethanol for 10 min, 70% ethanol for 5 min, 50% ethanol, distilled water for 5 min, and a final wash in 1 x PBS for 5 min. Antigen retrieval was performed in antigen retrieval buffer (10 mM Sodium Citrate, 0.05% tween-20, pH: 6) by heating the slides at 100 °C for 15 min. After antigen retrieval, permeabilization was performed with 0.04% Triton-X 100 in 1 X PBS (0.04% PBST) for 20 min followed by proper washing with 1 X PBS. Blocking was done by 3% BSA for 1 hr. Sequential addition of primary antibody was performed at 1:100 dilution at 4 °C overnight. Primary antibody was washed with 1 X PBS followed by counterstain with DAPI nuclear stain at 1 µg/ml concentration. Mounting was done with a drop of Vectashield.

## Histopathological studies

Formalin-fixed tissue samples (<5 mm thick) were washed under running water for 4–12 hr. Tissue dehydration was performed in ascending grades of alcohol (30%, 50%, 70%, 90%, 95%) followed by two washes in absolute alcohol, 1 hr each and two washes in xylene for 1 hr each. Paraffinization in melted paraffin wax (57–60°C) was performed three times sequentially for 1 hr each. FFPE (formalin-fixed paraffin-embedded) tissue blocks were prepared by placing tissue and melted paraffin in steel molds. Five-µm-thick sections of tissue were cut with the help of a microtome. Tissue sections were allowed to float on a water bath (45–50°C) and placed on a clean glass slide. Deparaffinization was performed by three sequential washes in xylene for 5 min each followed by hydration in descending grades of alcohol (absolute alcohol, 90% and 70%) followed by one wash in water, 5 min each. For H&E staining, slides were first stained with hematoxylin (it stains the nucleus blue) for 5–10 min followed by one wash with water. The slides were subsequently stained with Eosin (it stains the cytoplasm pink) for 5–10 min followed by one wash with absolute alcohol and one wash with xylene. Slides were mounted with coverslips followed by sealing with DPX. Images were acquired on Zeiss microscope.

## In vivo IL-17 neutralization experiment

IFNγ knockout C57BL/6 mice were challenged with 100–200 CFU with H37Rv via the aerosol route (as described previously). The anti-mouse IL-17 monoclonal Ab was administered via the intra-peritoneal route at a dose of 100 mg/mouse every 3–4 days starting one day before *M. tuberculosis* infection for the duration of the study, that is 6 weeks. Control groups received matched IgG isotypes. Six weeks post-*Mtb* infection, mice were treated with 50 mg/kg celecoxib through oral route for the duration of the experiments. Mice in all experimental groups were sacrificed at 6 weeks post-infection, according to the experimental timeline. The left half of the lungs and whole spleen of the mice were homogenized and serially diluted in 1 X PBS. Different dilutions were plated in duplicates on 7H11 agar plates and the colonies were counted after 21 days to enumerate the *Mtb* load. The middle lobe of the right lung was preserved in formalin for histopathology and IFA studies. The rest part of the right half lung

of the infected mice was used in the preparation of single-cell suspension. The single-cell suspension was divided into two parts and stained with different antibody cocktails. One part was used for FACS sorting and subsequent plating of sorted cells (i.e. neutrophils, macrophages, and mesenchymal stem cells), and the other half was used for FACS acquisition to enumerate Th17 cells inside the lungs. Sera was also collected to quantify IL-17 levels.

### IL-17/PGE2 ELISA

IL-17 levels and PGE2 levels in the lung homogenates of uninfected mice, *Mtb*-infected mice, *Mtb*-infected treated mice from various experimental groups were measured using ELISA kits procured from Abcam using the manufacturer's protocol.

### Study population (PTB subjects)

Participants were enrolled from the Effect of Diabetes on Tuberculosis Severity (EDOTS) study, a prospective cohort study conducted in Chennai, India. A total of 68 had unfavorable treatment outcomes, comprising treatment failure, relapse, or death. Inclusion criterion was new smear and culture-positive adults (age 18–75 years). Exclusion criteria were previous treatment for TB, drug-resistant TB, HIV positivity or taking immunosuppressive drugs, and >7 days of anti-TB treatment prior to enrollment. As part of our study protocol, HIV screening was done for all the study participants. The diagnosis of PTB was established by positive sputum culture on solid media with compatible chest X-ray. Anti-TB treatment (ATT) was provided by government TB clinics according to the Revised National Tuberculosis Control Programme standards in effect at that time. Participants were followed up monthly through the 6 months course of treatment and every three months thereafter till 1 year after treatment completion. Cure was defined as negative sputum cultures at months 5 and 6 of treatment.

Adverse treatment outcomes included treatment failure defined as positive sputum culture at months 5 or 6, all-cause mortality over the duration of treatment, or recurrent TB demonstrated by positive sputum smear and culture with compatible symptoms or chest X-ray within twelve months after initial cure. There was a total of 18 TB treatment failures, 16 deaths, and 34 TB recurrences. Case-control matching was based on age, gender, body mass index (BMI), and diabetic status. Peripheral blood was collected in heparinized tubes. Following centrifugation, plasma was collected and stored at –80 °C till further analysis. Baseline sample collection was performed at enrollment. The demographic and epidemiological data for this cohort have been previously reported (*Kumar et al., 2021*; *Kumar et al., 2020*).

### Human PTB multiplex assay

Circulating plasma levels of cytokines and chemokines in PTB patients were measured using the Luminex MAGPIX Multiplex Assay system (Bio-Rad, Hercules, CA). Luminex Human Magnetic Assay Kit 45 Plex (R&D Systems) was used to measure the cytokines and chemokine levels according to the manufacturer's protocol.

### Statistical analysis-RNAseq data

Transcriptomics analyses were done in R statistical software environment (v.4.1; *R Development Core Team, 2021*). Statistical analysis was done using swish methodology from fishpond (v.1.8; *Zhu et al., 2019*) and significantly differentially expressed genes (DEGs) were defined based on qvalue ≤0.1 and absolute $\log_2$ fold change ≥1. Enrichment analyses of the significant DEGs were done using gene set co-regulation analysis (geseca) from the fgsea (v1.18) R library (*Korotkevich et al., 2021*) and Gene Ontology (GO) Biological Processes (BP) and KEGG pathways from msigdbr database (v7.5.1). The significance of enrichment is defined by p.value≤0.05. A heatmap of the Luminex cytokine profile was generated with pheatmap (v.1.0.12) using mean z-scaled values and clustered based on Euclidean distance and Ward.D2 methodology.

## Acknowledgements

We thank the Tuberculosis Aerosol Challenge Facility (TACF), supported by the Department of Biotechnology, Govt of India, for supporting the BSL3 facility. We thank the staff of the Department of Clinical Research and the Department of Bacteriology, NIRT, for valuable assistance in bacterial

cultures and radiology and the staff of Prof. M Viswanathan Diabetes Research Center, RNTCP and Chennai corporation for valuable assistance in recruiting the patients for this study. Data in this manuscript were collected as part of the Regional Prospective Observational Research for Tuberculosis (RePORT) India Consortium. The work on this project was supported by Department of Biotechnology (BT/IC-06/003/91-Flagship program to DK), MDR-TB project from Department of Biotechnology (BT/PR23092/NER/95/589/2017, DK) and SR-National Bioscience Award from Department of Biotechnology (BT/HRD/NBA-NWB/39/2020-21 (4), DK). The research in the DK lab is also supported by the DBT-Wellcome Trust India Alliance Senior Fellowship (IA/S/17/1/503071) and extramural research grants from SERB (EMR/2016/005296 to DK). Priya Sharma is a recipient of a fellowship from the Indian Council of Medical Research (3/1/3/JRF-2017/HRD-LS/55193/79). Raman Deep Sharma is a recipient of a fellowship from the Department of Biotechnology (DBT/JRF/BET-18/I/2018/AL/160).

## Additional information

### Competing interests

Vinay Kumar Nandicoori, Dhiraj Kumar: Reviewing editor, eLife. The other authors declare that no competing interests exist.

### Funding

| Funder | Grant reference number | Author |
| --- | --- | --- |
| Department of Biotechnology, Ministry of Science and Technology, India | BT/IC-06/003/91 | Dhiraj Kumar |
| Department of Biotechnology, Ministry of Science and Technology, India | BT/PR23092/NER/95/589/2017 | Dhiraj Kumar |
| Department of Biotechnology, Ministry of Science and Technology, India | BT/HRD/NBA-NWB/39/2020-21 | Dhiraj Kumar |
| Wellcome Trust DBT India Alliance | IA/S/17/1/503071 | Dhiraj Kumar |
| Science and Engineering Research Board | EMR/2016/005296 | Dhiraj Kumar |
| Indian Council of Medical Research | 3/1/3/JRF-2017/HRD-LS/55193/79 | Priya Sharma |
| Department of Biotechnology | DBT/JRF/BET-18/I/2018/AL/160 | Raman Deep Sharma |

The funders had no role in study design, data collection and interpretation, or the decision to submit the work for publication. For the purpose of Open Access, the authors have applied a CC BY public copyright license to any Author Accepted Manuscript version arising from this submission.

### Author contributions

Priya Sharma, Data curation, Formal analysis, Validation, Investigation, Visualization, Methodology, Writing – original draft, Writing – review and editing; Raman Deep Sharma, Investigation, Methodology; Binayak Sarkar, Lalita Mehra, Investigation, Visualization; Varnika Panwar, Data curation, Investigation; Mrinmoy Das, Data curation, Software, Investigation, Visualization; Lakshya Veer Singh, Neharika Jain, Aditya Rathee, Shilpa Sharma, Methodology; Shivam Chaturvedi, Resources, Methodology; Shihui Foo, Data curation, Methodology; Andrea Lee, Data curation, Software, Methodology; Pavan Kumar N, Resources, Data curation; Prasenjit Das, Investigation, Visualization, Methodology, Writing – original draft; Vijay Viswanathan, Hardy Kornfeld, Subash Babu, Data curation; Shanshan W

Howland, Formal analysis; Vinay Kumar Nandicoori, Resources, Formal analysis, Investigation, Writing – original draft; Amit Singhal, Resources, Data curation, Formal analysis, Funding acquisition, Investigation, Methodology, Writing – original draft, Writing – review and editing; Dhiraj Kumar, Conceptualization, Resources, Formal analysis, Supervision, Funding acquisition, Visualization, Methodology, Writing – original draft, Writing – review and editing

### Author ORCIDs
Priya Sharma ⬤ https://orcid.org/0009-0005-4553-1512
Shivam Chaturvedi ⬤ https://orcid.org/0000-0002-6520-1191
Hardy Kornfeld ⬤ https://orcid.org/0000-0002-8970-7306
Vinay Kumar Nandicoori ⬤ https://orcid.org/0000-0002-5682-4178
Dhiraj Kumar ⬤ https://orcid.org/0000-0001-7578-2930

### Ethics
ITB biopsy samples and control samples were obtained from the Department of Pathology, AIIMS, New Delhi. These biopsy samples were taken from the patients for diagnostic purposes after taking their written informed consent. The use of the archived leftover biopsy samples was approved by the institute's EC of AIIMS (Ref no. IEC-304/02-06-2017), and ICGEB (Ref no. ICGEB/IEC/2017/06-verII and ICGEB/IEC/2016/03) and the same were accessed according to the institutional guidelines. The study on human PTB subjects was approved by the Ethics Committees of the Prof. M. Viswanathan Diabetes Research Center (ECR/51/INST/TN/ 2013/MVDRC/01) and NIRT (NIRT-INo:2014004). Informed written consent was obtained from all participants.

Animal experiments were approved by Institutional Animal Ethics Committee, ICGEB (ICGEB/IAEC/280718/CI-14), (ICGEB/IAEC/07032020/CI-18), and (ICGEB/IAEC/18092021/CI-16).

Reviewer #2 (Public review): https://doi.org/10.7554/eLife.100966.3.sa1
Reviewer #3 (Public review): https://doi.org/10.7554/eLife.100966.3.sa2
Author response https://doi.org/10.7554/eLife.100966.3.sa3

---

## Additional files

### Supplementary files
Supplementary file 1. Differentially expressed genes (DEGs) in PMNs identified between each pairwise comparison.

Supplementary file 2. Differentially regulated genes in Ly6G$^+$Gra from unvaccinated, uninfected versus BCG-vaccinated, uninfected animals.

Supplementary file 3. Gene-set co-regulation analysis of the DEGs with GO-BP in PMNs among the Uninfected control group, *Mtb*-infected group, and BCG-vaccinated *Mtb*-infected group.

Supplementary file 4. Gene-set co-regulation analysis of the DEGs with KEGG pathways in PMNs among the Uninfected control group, *Mtb*-infected group, and BCG-vaccinated *Mtb*-infected group.

MDAR checklist

### Data availability
The raw RNAseq data used for this study is available at the GEO database with the accession number: GSE238265.

The following dataset was generated:

| Author(s) | Year | Dataset title | Dataset URL | Database and Identifier |
| --- | --- | --- | --- | --- |
| Sharma P, Das M, Kumar D, Singhal A | 2025 | An IL-17-Ly6G+Polymorphonuclear Neutrophil (PMN) axis limits protective host responses against tuberculosis | https://www.ncbi.nlm.nih.gov/geo/query/acc.cgi?acc=GSE238265 | NCBI Gene Expression Omnibus, GSE238265 |

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
