## [Editor Report · eLife Assessment]

This **valuable** study examines the role of IL17-producing Ly6G PMNs as a reservoir for *Mycobacterium tuberculosis* to evade host killing activated by BCG immunisation. The authors provide **solid** data reporting that IL17-producing polymorphonuclear neutrophils harbour a significant bacterial load in both wild-type and IFNg-/- mice and that targeting IL17 and Cox2 improved disease outcomes whilst enhancing BCG efficacy. The specific contribution of neutrophil-derived IL-17 to disease pathogenesis remains to be definitively established through direct demonstration of IL-17 production by neutrophils and targeted depletion studies.

---

## [Referee Report · Reviewer #2 (Public review)]

Summary:

In this study, Sharma et al. demonstrated that Ly6G+ granulocytes (Gra cells) serve as the primary reservoirs for intracellular Mtb in infected wild-type mice and that excessive infiltration of these cells is associated with severe bacteremia in genetically susceptible IFNγ-/- mice. Notably, neutralizing IL-17 or inhibiting COX2 reversed the excessive infiltration of Ly6G+Gra cells, mitigated the associated pathology, and improved survival in these susceptible mice. Additionally, Ly6G+Gra cells were identified as a major source of IL-17 in both wild-type and IFNγ-/- mice. Inhibition of RORγt or COX2 further reduced the intracellular bacterial burden in Ly6G+Gra cells and improved lung pathology.

Of particular interest, COX2 inhibition in wild-type mice also enhanced the efficacy of the BCG vaccine by targeting the Ly6G+Gra-resident Mtb population.

Strengths:

The experimental results showing improved BCG-mediated protective immunity through targeting IL-17-producing Ly6G+ cells and COX2 are compelling and will likely generate significant interest in the field. Overall, this study presents important findings, suggesting that the IL-17-COX2 axis could be a critical target for designing innovative vaccination strategies for TB.

Comments on revisions:

This article is of significant interest for the research field. In the revised version of the manuscript the authors have addressed the concerns raised during initial review. I do not have further concerns.

---

## [Referee Report · Reviewer #3 (Public review)]

Summary:

The authors examine how distinct cellular environments differentially control Mtb following BCG vaccination. The key findings are that IL17 producing PMNs harbor a significant Mtb load in both wild type and IFNg-/- mice. Targeting IL17, Cox2, and Rorgt, improved disease in combination but not alone and enhances BCG efficacy over 12 weeks and neutrophils/IL17 are associated with treatment failure in humans. The authors suggest that targeting these pathways, especially in MSMD patients may improve disease outcomes.

Strengths:

The experimental approach is generally sound and consists of low dose aerosol infections with distinct readouts including cell sorting followed by CFU, histopathology and RNA sequencing analysis. By combining genetic approaches and chemical/antibody treatments, the authors can probe these pathways effectively.

Understanding how distinct inflammatory pathways contribute to control or worsen Mtb disease is important and thus, the results will be of great interest to the Mtb field.

Uncovering a neutrophil population that is refractory to BCG-mediated control can help to better define key markers for vaccine efficacy

Weaknesses:

Several of the key findings in mice have previously been shown (albeit with less sophisticated experimentation) and human disease and neutrophils are well described - thus the real new finding is how intracellular Mtb in neutrophils are more refractory to BCG-mediated control and modulating IL17 and inflammation can alter this.

There is a lack of direct evidence that the neutrophils are producing IL-17 or showing that specifically removing IL17 neutrophils has an effect on disease. Thus, many of these data are correlative, or have modest phenotypes. For example if blocking IL17 or alone does not impact disease alone the conclusion that these IL17+ neutrophils limits protection as noted in the title is is not fully supported. The inhibitors used are not cell-type specific.

---

## [Author Response]

The following is the authors’ response to the original reviews.

**Reviewer #1 (Public review):**
Summary:Recruitment of neutrophils to the lungs is known to drive susceptibility to infection with *M. tuberculosis*. In this study, the authors present data in support of the hypothesis that neutrophil production of the cytokine IL-17 underlies the detrimental effect of neutrophils on disease. They claim that neutrophils harbor a large fraction of Mtb during infection, and are a major source of IL-17. To explore the effects of blocking IL-17 signaling during primary infection, they use IL-17 blocking antibodies, SR221 (an inverse agonist of Th17 differentiation), and celecoxib, which they claim blocks Th17 differentiation, and observe modest improvements in bacterial burdens in both WT and IFN-γ deficient mice using the combination of IL-17 blockade with celecoxib during primary infection. Celecoxib enhances control of infection after BCG vaccination.

Thank you for the summary.

Strengths:The most novel finding in the paper is that treatment with celecoxib significantly enhances control of infection in BCG-vaccinated mice that have been challenged with Mtb. It was already known that NSAID treatments can improve primary infection with Mtb.

Thank you.

Weaknesses:The major claim of the manuscript - that neutrophils produce IL-17 that is detrimental to the host - is not strongly supported by the data. Data demonstrating neutrophil production of IL17 lacks rigor.

Our response: Neutrophil production of IL-17 is supported by two independent methods/ techniques in the current version:

(1) Through Flow cytometry- a large fraction of Ly6G^+^CD11b^+^ cells from the lungs of Mtb-infected mice were also positive for IL-17 (Fig. 3C).

(2) IFA co-staining of Ly6G ^+^ cells with IL-17 in the lung sections from Mtb-infected mice (Fig. 3 E_G and Fig. 4H, Fig. 5I). For most of these IFA data, we provide quantified plots to show IL17^+^Ly6G^+^ cells.

(3) Most importantly, conditions that inhibited IL-17 levels and controlled infection also showed a decline in IL-17 staining in Ly6G^+^ cells.

Our efforts on IL-17 ELISPOT assay were not very successful and it needs further standardization.

Several independent publications support the production of IL-17 by neutrophils (Li et al. 2010; Katayama et al. 2013; Lin et al. 2011). For example, neutrophils have been identified as a source of IL-17 in human psoriatic lesions (Lin et al. 2011), in neuroinflammation induced by traumatic brain injury (Xu et al. 2023) and in several mouse models of infectious and autoimmune inflammation (Ferretti et al. 2003; Hoshino et al. 2008) (Li et al. 2010).

The experiments examining the effects of inhibitors of IL-17 on the outcome of infection are very difficult to interpret. First, treatment with IL-17 inhibitors alone has no impact on bacterial burdens in the lung, either in WT or IFN-γ KO mice. This suggests that IL-17 does not play a detrimental role during infection. Modest effects are observed using the combination of IL-17 blocking drugs and celecoxib, however, the interpretation of these results mechanistically is complicated. Celecoxib is not a specific inhibitor of Th17. Indeed, it affects levels of PGE2, which is known to have numerous impacts on Mtb infection separate from any effect on IL-17 production, as well as other eicosanoids.

The reviewer correctly says that Celecoxib is not a specific inhibitor of Th17. However, COX2 inhibition does have an effect on IL-17 levels, and numerous reports support this observation (Paulissen et al. 2013; Napolitani et al. 2009; Lemos et al. 2009).

(1) The detrimental role of IL-17 is obvious in the IFNγ KO experiment, where IL-17 neutralization led to a significant improvement in the lung pathology.

(2) In the highly susceptible IFNγ KO mice, IL-17 neutralization alone extended the survival of mice by ~10 days.

(3) IL-17 production independent of IL-23 is known to require PGE2 (Paulissen et al. 2013; Polese et al. 2021). In either WT or IFNγ KO mice, in contrast to IL-17 levels, we observed a decline in IL-23 levels. The PGE2 dependence of IL-17 production is obvious in the WT mice, where celecoxib abrogated IL-17 production.

(4) While deciding the impact of celecoxib or IL17 inhibition, looking at the cumulative readout of lung CFU, spleen CFU, Ly6G^+^ cell recruitment, Ly6G^+^ cell-resident Mtb pool and overall pathology, the effects are quite significant.

(5) Finally, in the revised manuscript, we provide additional results on the effect of SR2211 in BCG-vaccinated animals. It shows the direct impact of IL-17 inhibition on the BCG vaccine efficacy in WT mice.

Finally, the human data simply demonstrates that neutrophils and IL-17 both are higher in patients who experience relapse after treatment for TB, which is expected and does not support their specific hypothesis.

We disagree with the above statement. It also contradicts reviewers’ own assessments in one of the comments below, where a protective role of IL-17 is referred to. The literature lacks consensus in terms of a protective or pathological role of IL-17 in TB. Therefore, it was not expected to see higher IL-17 in patients who experienced relapse, death, or failed treatment outcomes. We do not have evidence from human subjects whether neutrophil-derived IL-17 has a similar pathological role as observed in mice. However, higher IL-17 in failed outcome cases confirm the central theme that IL-17 is pathological in both human and mouse models.

The use of genetic ablation of IL-17 production specifically in neutrophils and/or IL-17R in mice would greatly enhance the rigor of this study.

The reviewer’s point is well-taken. Having a genetic ablation of IL-17 production, specifically in the neutrophils, would be excellent. At present, however, we lack this resource. For the revised manuscript, we include the data with SR2211, a direct inhibitor of RORgt and, therefore, IL-17, in BCG-vaccinated mice.

The authors do not address the fact that numerous studies have shown that IL-17 has a protective effect in the mouse model of TB in the context of vaccination.

Yes, there are a few articles that talk about the protective effect of IL-17 in the mouse model of TB in the context of vaccination (Khader et al. 2007; Desel et al. 2011; Choi et al. 2020). This part was discussed in the original manuscript (in the Introduction section). For the revised manuscript, we also provide results from the experiment where we blocked IL-17 production by inhibiting RORgt using SR2211 in BCG-vaccinated mice. The results clearly show IL-17 as a negative regulator of BCG-mediated protective immunity. We believe some of the reasons for the observed differences could be (1) in our study, we analysed IL-17 levels in the lung homogenates at late phases of infection, and (2) most published studies rely on ex vivo stimulation of immune cells to measure cytokine production, whereas we actually measured the cytokine levels in the lung homogenates. We will elaborate on these points in the revised version.

Finally, whether and how many times each animal experiment was repeated is unclear.

We provide the details of the number of experiments in the revised version. Briefly, the BCG vaccination experiment (Figure 1) and BCG vaccination with Celecoxib treatment experiment (Figure 6) were performed twice and thrice, respectively. The IL-17 neutralization experiment (Figure 4) and the SR2211 treatment experiment (Figure 5) were done once. We will add another SR2211 experiment data in the revised version.

**Reviewer #2 (Public review):**
Summary:In this study, Sharma et al. demonstrated that Ly6G+ granulocytes (Gra cells) serve as the primary reservoirs for intracellular Mtb in infected wild-type mice and that excessive infiltration of these cells is associated with severe bacteremia in genetically susceptible IFNγ/- mice. Notably, neutralizing IL-17 or inhibiting COX2 reversed the excessive infiltration of Ly6G+Gra cells, mitigated the associated pathology, and improved survival in these susceptible mice. Additionally, Ly6G+Gra cells were identified as a major source of IL-17 in both wild-type and IFNγ-/- mice. Inhibition of RORγt or COX2 further reduced the intracellular bacterial burden in Ly6G+Gra cells and improved lung pathology.Of particular interest, COX2 inhibition in wild-type mice also enhanced the efficacy of the BCG vaccine by targeting the Ly6G+Gra-resident Mtb population.

Thank you for the summary.

Strengths:The experimental results showing improved BCG-mediated protective immunity through targeting IL-17-producing Ly6G+ cells and COX2 are compelling and will likely generate significant interest in the field. Overall, this study presents important findings, suggesting that the IL-17-COX2 axis could be a critical target for designing innovative vaccination strategies for TB.

Thank you for highlighting the overall strengths of the study.

Weaknesses:However, I have the following concerns regarding some of the conclusions drawn from the experiments, which require additional experimental evidence to support and strengthen the overall study.Major Concerns:(1) Ly6G+ Granulocytes as a Source of IL-17: The authors assert that Ly6G+ granulocytes are the major source of IL17 in wild-type and IFN-γ KO mice based on colocalization studies of Ly6G and IL-17. In Figure 3D, they report approximately 500 Ly6G+ cells expressing IL-17 in the Mtb-infected WT lung. Are these low numbers sufficient to drive inflammatory pathology? Additionally, have the authors evaluated these numbers in IFN-γ KO mice?

Thank you for pointing out the numbers in Fig. 3D It was our oversight to label the axis as No. of. For the observation that Ly6G^+^ Gra are the major source of IL-17 in TB, we have used two separate strategies- (a) IFA and (b) FACS IL17^+^ Ly6G^+^ Gra/lung. For this data, only a part of the lung was used. For the revised manuscript, we provide the number of these cells at the whole lung level from Mtb-infected WT mice. Unfortunately, we did not evaluate these numbers in IFN-γ KO mice through FACS..

Our efforts to perform the IL-17 ELISpot assay on the sorted Ly6G^+^Gra from the lungs of Mtbinfected WT mice were unsuccessful. However, we provide a quantified representation of IFA of the tissue sections to stress upon the role of Ly6G^+^ cells in IL-17 production in TB pathogenesis.

(2) Role of IL-17-Producing Ly6G Granulocytes in Pathology: The authors suggest that IL-17producing Ly6G granulocytes drive pathology in WT and IFN-γ KO mice. However, the data presented only demonstrate an association between IL-17^+^ Ly6G cells and disease pathology. To strengthen their conclusion, the authors should deplete neutrophils in these mice to show that IL-17 expression, and consequently the pathology, is reduced.

Thank you for this suggestion. Neutrophil depletion studies in TB remain inconclusive. In some studies, neutrophil depletion helps the pathogen (Rankin et al. 2022; Pedrosa et al. 2000; Appelberg et al. 1995), and in others, it helps the host (Lovewell et al. 2021; Mishra et al. 2017). One reason for this variability is the stage of infection when neutrophil depletion was done. However, another crucial factor is the heterogeneity in the neutrophil population. There are reports that suggest neutrophil subtypes with protective versus pathological trajectories (Nwongbouwoh Muefong et al. 2022; Lyadova 2017; Hellebrekers, Vrisekoop, and Koenderman 2018; Leliefeld et al. 2018). Depleting the entire population using anti-Ly6G could impact this heterogeneity and may impact the inferences drawn.

A better approach would be to characterise this heterogeneous population, efforts towards which could be part of a separate study. Another direct approach could be Ly6G^+^-specific deletion of IL-17 function as part of a separate study.

For the revised manuscript, we provide results from the SR2211 experiment in BCG-vaccinated mice and other results to show the role of IL-17-producing Ly6G^+^ Gra in TB pathology.

(3) IL-17 Secretion by Mtb-Infected Neutrophils: Do Mtb-infected neutrophils secrete IL-17 into the supernatants? This would serve as confirmation of neutrophil-derived IL-17. Additionally, are Ly6G^+^ cells producing IL-17 and serving as pathogenic agents exclusively in vivo? The authors should provide comments on this.

Secretion of IL-17 by Mtb-infected neutrophils in vitro has been reported earlier (Hu et al. 2017). Our efforts to do a neutrophil IL-17 ELISPOT assay were not successful, and we are still standardising it. Whether there are a few neutrophil roles exclusively seen under in vivo conditions is an interesting proposition.

(4) Characterization of IL-17-Producing Ly6G+ Granulocytes: Are the IL-17-producing Ly6G+ granulocytes a mixed population of neutrophils and eosinophils, or are they exclusively neutrophils? Sorting these cells followed by Giemsa or eosin staining could clarify this.

This is a very important point. While usually eosinophils do not express Ly6G markers in laboratory mice, under specific contexts, including infections, eosinophils can express Ly6G. Since we have not characterized these potential Ly6G^+^ sub-populations, that is one of the reasons we refer to the cell types as Ly6G^+^ granulocytes, which do not exclude Ly6G^+^ eosinophils. A detailed characterization of these subsets could be taken up as a separate study.

**Reviewer #3 (Public review):**
Summary:The authors examine how distinct cellular environments differentially control Mtb following BCG vaccination. The key findings are that IL17-producing PMNs harbor a significant Mtb load in both wild-type and IFNg^-/-^ mice. Targeting IL17 and Cox2 improved disease and enhanced BCG efficacy over 12 weeks and neutrophils/IL17 are associated with treatment failure in humans. The authors suggest that targeting these pathways, especially in MSMD patients may improve disease outcomes.

Thank you.

Strengths:The experimental approach is generally sound and consists of low-dose aerosol infections with distinct readouts including cell sorting followed by CFU, histopathology, and RNA sequencing analysis. By combining genetic approaches and chemical/antibody treatments, the authors can probe these pathways effectively.Understanding how distinct inflammatory pathways contribute to control or worsen Mtb disease is important and thus, the results will be of great interest to the Mtb field

Thank you.

Weaknesses:A major limitation of the current study is overlooking the role of non-hematopoietic cells in the IFNg/IL17/neutrophil response. Chimera studies from Ernst and colleagues (Desvignes and Ernst 2009) previously described this IDO-dependent pathway following the loss of IFNg through an increased IL17 response. This study is not cited nor discussed even though it may alter the interpretation of several experiments.

Thank you for pointing out this earlier study, which we concede, we missed discussing. We disagree on the point that results from that study may alter the interpretation of several experiments in our study. On the contrary, the main observation that loss of IFNγ causes severe IL-17 levels is aligned in both studies.

IDO1 is known to alter T-helper cell differentiation towards Tregs and away from Th17 (Baban et al. 2009). It is absolutely feasible for the non-hematopoietic cells to regulate these events. However, that does not rule out the neutrophil production of IL-17 and the downstream pathological effect shown in this study. We have discussed and cited this study in the revised manuscript.

Several of the key findings in mice have previously been shown (albeit with less sophisticated experimentation) and human disease and neutrophils are well described - thus the real new finding is how intracellular Mtb in neutrophils are more refractory to BCG-mediated control. However, given there are already high levels of Mtb in PMNs compared to other cell types, and there is a decrease in intracellular Mtb in PMNs following BCG immunization the strength of this finding is a bit limited.

The reviewer’s interpretation of the BCG-refractory Mtb population in the neutrophil is interesting. The reviewer is right that neutrophils had a higher intracellular Mtb burden, which decreased in the BCG-vaccinated animals. Thus, on that account, the reviewer rightly mentions that BCG is able to control Mtb even in neutrophils. However, BCG almost clears intracellular burden from other cell types analysed, and therefore, the remnant pool of intracellular Mtb in the lungs of BCG-vaccinated animals could be mostly those present in the neutrophils. This is a substantial novel development in the field and attracts focus towards innate immune cells for vaccine efficacy.

References:

Appelberg, R., A. G. Castro, S. Gomes, J. Pedrosa, and M. T. Silva. 1995. 'SuscepBbility of beige mice to Mycobacterium avium: role of neutrophils', Infect Immun, 63: 3381-7.

Baban, B., P. R. Chandler, M. D. Sharma, J. Pihkala, P. A. Koni, D. H. Munn, and A. L. Mellor. 2009. 'IDO acBvates regulatory T cells and blocks their conversion into Th17-like T cells', J Immunol, 183: 2475-83.

Choi, H. G., K. W. Kwon, S. Choi, Y. W. Back, H. S. Park, S. M. Kang, E. Choi, S. J. Shin, and H. J. Kim. 2020. 'AnBgen-Specific IFN-gamma/IL-17-Co-Producing CD4(+) T-Cells Are the Determinants for ProtecBve Efficacy of Tuberculosis Subunit Vaccine', Vaccines (Basel), 8.

Cruz, A., A. G. Fraga, J. J. Fountain, J. Rangel-Moreno, E. Torrado, M. Saraiva, D. R. Pereira, T. D. Randall, J. Pedrosa, A. M. Cooper, and A. G. Castro. 2010. 'Pathological role of interleukin 17 in mice subjected to repeated BCG vaccinaBon afer infecBon with *Mycobacterium tuberculosis*', J Exp Med, 207: 1609-16.

Desel, C., A. Dorhoi, S. Bandermann, L. Grode, B. Eisele, and S. H. Kaufmann. 2011. 'Recombinant BCG DeltaureC hly+ induces superior protecBon over parental BCG by sBmulaBng a balanced combinaBon of type 1 and type 17 cytokine responses', J Infect Dis, 204: 1573-84.

Desvignes, L., and J. D. Ernst. 2009. 'Interferon-gamma-responsive nonhematopoieBc cells regulate the immune response to *Mycobacterium tuberculosis*', Immunity, 31: 974-85.

Ferreg, S., O. Bonneau, G. R. Dubois, C. E. Jones, and A. Trifilieff. 2003. 'IL-17, produced by lymphocytes and neutrophils, is necessary for lipopolysaccharide-induced airway neutrophilia: IL-15 as a possible trigger', J Immunol, 170: 2106-12.

Hellebrekers, P., N. Vrisekoop, and L. Koenderman. 2018. 'Neutrophil phenotypes in health and disease', Eur J Clin Invest, 48 Suppl 2: e12943.

Hoshino, A., T. Nagao, N. Nagi-Miura, N. Ohno, M. Yasuhara, K. Yamamoto, T. Nakayama, and K. Suzuki. 2008. 'MPO-ANCA induces IL-17 producBon by acBvated neutrophils in vitro via classical complement pathway-dependent manner', J Autoimmun, 31: 79-89.

Hu, S., W. He, X. Du, J. Yang, Q. Wen, X. P. Zhong, and L. Ma. 2017. 'IL-17 ProducBon of Neutrophils Enhances AnBbacteria Ability but Promotes ArthriBs Development During *Mycobacterium tuberculosis* InfecBon', EBioMedicine, 23: 88-99.

Hult, C., J. T. Magla, H. P. Gideon, J. J. Linderman, and D. E. Kirschner. 2021. 'Neutrophil Dynamics Affect *Mycobacterium tuberculosis* Granuloma Outcomes and DisseminaBon', Front Immunol, 12: 712457.

Katayama, M., K. Ohmura, N. Yukawa, C. Terao, M. Hashimoto, H. Yoshifuji, D. Kawabata, T. Fujii, Y. Iwakura, and T. Mimori. 2013. 'Neutrophils are essenBal as a source of IL-17 in the effector phase of arthriBs', PLoS One, 8: e62231.

Khader, S. A., G. K. Bell, J. E. Pearl, J. J. Fountain, J. Rangel-Moreno, G. E. Cilley, F. Shen, S. M. Eaton, S. L. Gaffen, S. L. Swain, R. M. Locksley, L. Haynes, T. D. Randall, and A. M. Cooper. 2007. 'IL-23 and IL-17 in the establishment of protecBve pulmonary CD4+ T cell responses afer vaccinaBon and during *Mycobacterium tuberculosis* challenge', Nat Immunol, 8: 369-77.

Leliefeld, P. H. C., J. Pillay, N. Vrisekoop, M. Heeres, T. Tak, M. Kox, S. H. M. Rooijakkers, T. W. Kuijpers, P. Pickkers, L. P. H. Leenen, and L. Koenderman. 2018. 'DifferenBal anBbacterial control by neutrophil subsets', Blood Adv, 2: 1344-55.

Lemos, H. P., R. Grespan, S. M. Vieira, T. M. Cunha, W. A. Verri, Jr., K. S. Fernandes, F. O. Souto, I. B. McInnes, S. H. Ferreira, F. Y. Liew, and F. Q. Cunha. 2009. 'Prostaglandin mediates IL-23/IL-17induced neutrophil migraBon in inflammaBon by inhibiBng IL-12 and IFNgamma producBon', Proc Natl Acad Sci U S A, 106: 5954-9.

Li, L., L. Huang, A. L. Vergis, H. Ye, A. Bajwa, V. Narayan, R. M. Strieter, D. L. Rosin, and M. D. Okusa. 2010. 'IL-17 produced by neutrophils regulates IFN-gamma-mediated neutrophil migraBon in mouse kidney ischemia-reperfusion injury', J Clin Invest, 120: 331-42.

Lin, A. M., C. J. Rubin, R. Khandpur, J. Y. Wang, M. Riblen, S. Yalavarthi, E. C. Villanueva, P. Shah, M. J. Kaplan, and A. T. Bruce. 2011. 'Mast cells and neutrophils release IL-17 through extracellular trap formaBon in psoriasis', J Immunol, 187: 490-500.

Lovewell, R. R., C. E. Baer, B. B. Mishra, C. M. Smith, and C. M. Sasseg. 2021. 'Granulocytes act as a niche for *Mycobacterium tuberculosis* growth', Mucosal Immunol, 14: 229-41.

Lyadova, I. V. 2017. 'Neutrophils in Tuberculosis: Heterogeneity Shapes the Way?', Mediators Inflamm, 2017: 8619307.

Mishra, B. B., R. R. Lovewell, A. J. Olive, G. Zhang, W. Wang, E. Eugenin, C. M. Smith, J. Y. Phuah, J. E. Long, M. L. Dubuke, S. G. Palace, J. D. Goguen, R. E. Baker, S. Nambi, R. Mishra, M. G. Booty, C. E. Baer, S. A. Shaffer, V. Dartois, B. A. McCormick, X. Chen, and C. M. Sasseg. 2017. 'Nitric oxide prevents a pathogen-permissive granulocyBc inflammaBon during tuberculosis', Nat Microbiol, 2: 17072.

Napolitani, G., E. V. Acosta-Rodriguez, A. Lanzavecchia, and F. Sallusto. 2009. 'Prostaglandin E2 enhances Th17 responses via modulaBon of IL-17 and IFN-gamma producBon by memory CD4+ T cells', Eur J Immunol, 39: 1301-12.

Nwongbouwoh Muefong, C., O. Owolabi, S. Donkor, S. Charalambous, A. Bakuli, A. Rachow, C. Geldmacher, and J. S. Sutherland. 2022. 'Neutrophils Contribute to Severity of Tuberculosis

Pathology and Recovery From Lung Damage Pre- and Posnreatment', Clin Infect Dis, 74: 175766.

Paulissen, S. M., J. P. van Hamburg, N. Davelaar, P. S. Asmawidjaja, J. M. Hazes, and E. Lubberts. 2013. 'Synovial fibroblasts directly induce Th17 pathogenicity via the cyclooxygenase/prostaglandin E2 pathway, independent of IL-23', J Immunol, 191: 1364-72.

Pedrosa, J., B. M. Saunders, R. Appelberg, I. M. Orme, M. T. Silva, and A. M. Cooper. 2000. 'Neutrophils play a protecBve nonphagocyBc role in systemic *Mycobacterium tuberculosis* infecBon of mice', Infect Immun, 68: 577-83.

Polese, B., B. Thurairajah, H. Zhang, C. L. Soo, C. A. McMahon, G. Fontes, S. N. A. Hussain, V. Abadie, and I. L. King. 2021. 'Prostaglandin E(2) amplifies IL-17 producBon by gammadelta T cells during barrier inflammaBon', Cell Rep, 36: 109456.

Rankin, A. N., S. V. Hendrix, S. K. Naik, and C. L. Stallings. 2022. 'Exploring the Role of Low-Density Neutrophils During *Mycobacterium tuberculosis* InfecBon', Front Cell Infect Microbiol, 12: 901590.

Xu, X. J., Q. Q. Ge, M. S. Yang, Y. Zhuang, B. Zhang, J. Q. Dong, F. Niu, H. Li, and B. Y. Liu. 2023. 'Neutrophil-derived interleukin-17A parBcipates in neuroinflammaBon induced by traumaBc brain injury', Neural Regen Res, 18: 1046-51.

**Reviewer #1 (Recommendations for the authors):**
All figures: Clear information about the number of repeat experiments for each figure must be included.

We have provided the details of the number of repeat experiments in the revised version.

Figure 1: The claim that neutrophils are a dominant cell type infected during Mtb infection of the lungs is undermined by the limited number of markers used to identify cell types. The gating strategy used to initially identify what cells are infected with Mtb divided cells into three categories; granulocytes (Ly6G^+^ Cd11b^+^), CD64+MerTK+ macrophages, or Sca1+CD90.1+CD73+ (mesenchymal stem cells). This strategy leaves out monocyte populations that have been shown to be the dominant infected cells in other strategies (most recently, PMID: 36711606).

Thank you for this important point. We agree that we did not assess the infected monocyte population, specifically the Cd11c^+^ population. Both CD11c^Hi^ and CD11c^Lo^ monocyte cells appear to be important for Mtb infection, in different studies (Lee et al., 2020), (Zheng et al., 2024). Therefore, leaving out the CD11c^+^ population in our assays was a conscious decision to ensure the clarity of the cell types being studied.

In addition, substantial evidence from multiple studies indicates that Ly6G⁺ granulocytes constitute the predominant infected population in the Mtb-infected lungs of both mice and humans (Lovewell et al., 2021) (Eum et al., 2010). While monocytes may contribute to Mtb infection dynamics, our findings align with a growing body of research emphasizing the significant role of neutrophils as a dominant infected cell type in the lungs during TB pathology.

Figure 1: Putting the data from separate panels together, it appears that very few bacteria are isolated from the three cell types in the lung, suggesting there may be some loss in the preparation steps. Why is the total sorted CFU from neutrophils, macrophages, and MSCs so low, <400 bacteria total, when the absolute CFU is so high? Is it because only a fraction of the lung is being sorted/plated?

Yes, only a fraction of the lung was used for cell sorting and subsequent plating. The CFU plating from sorted cells also does not account for any bacteria growing extracellularly.

Figure 3C: It is difficult to ascertain whether the gating on IL-17^+^ cells is accurately identifying IL-17 producing cells. It is surprising, based on other published work, that the authors claim that almost half of CD45+CD11b-Ly6G- cells produce IL-17 in WT mice. It would be informative to show cell type-specific production of IL-17 in both WT and IFN-γ KO mice for comparison with the literature. Unstained/isotype controls for IL-17 staining should be shown. With this in mind, it is difficult to interpret the authors' claim that 80% of neutrophils produce IL-17.

Thank you for the points above. We do agree that we were surprised to see ~50% of CD45^+^ CD11b^-^Ly6G^-^ cells producing IL-17. We have now done multiple experiments to confirm that this number is actually less than 1% (~90 cells) in the uninfected mice and less than 4% (~4000) in the Mtb-infected mice.

Neutrophil-derived IL-17 production in Mtb-infected lungs is supported by two independent techniques in our current study: Flow Cytometry and Immunofluorescence assay. While Neutrophil production of IL-17 is rarely studied in the context of TB, in several other settings it has been widely reported (Gonzalez-Orozco et al., 2019; Li et al., 2010; Ramirez-Velazquez et al., 2013). We consistently get >60% IL-17 positive cells in the CD11b^+^ Ly6G^+^ population, specifically in the infected samples.

To specifically address the reviewer’s concerns, we have now used an isotype control for IL17 staining and show the specificity of IL-17A antibody binding. The Author response image 1 is from the uninfected mice, 8 weeks age.

Unfortunately, our efforts to establish an IL-17 ELISPOT assay from neutrophils were not very successful and need further standardisation. The new results are included in Fig. 3C-D and Fig. S2F-G in the revised manuscript.

Figure 3 D-H. Quantification of immunofluorescence microscopy should be provided.

In the revised manuscript, we provide the quantification of IFA results.

Figure 4: Effects on neutrophil numbers in IFN-γ Kos do not correlate with CFU reductions, suggesting there may be a neutrophilindependent mechanism.

In the IFN-γ KO, we agree that the effect was less than dramatic. The immune dysfunction in the IFN-γ KO mice is too severe to see a strong reversal in the phenotype through interventions.

While we do not rule out any neutrophil-independent mechanism, in the context of following observations, neutrophil-dependent mechanisms certainly appear to play an important role-

(a) Improved pathology and survival upon IL-17 neutralization, which further improves with the inclusion of celecoxib.

(b) Loss of IL17^+^-Ly6G^+^ cells upon IL-17 neutralization, which is further exacerbated when combined with celecoxib.

(c) Significant reduction in PMN number (shown by FACS) without any major impact on Th17 cell population upon IL-17 neutralization.

Finally, we believe some of the observations may become stronger once we characterize the specific sub-population among the Ly6G+ cells that correlates with pathology. For example, as shown in Figure 4I, FACS analysis of the Ly6G^⁺^ cell population in Mtb-infected IFNγ^⁻/⁻^ mice revealed a substantial subset of CD11b^mid^ Ly6G^ʰⁱ^ cells, indicative of an immature neutrophil population (Scapini et al., 2016). Efforts are currently underway to identify these important subpopulations.

Figure 4: Differences observed in the spleen cannot be connected to dissemination per se but instead could be a result of enhanced immune control in the spleen.

Thank you for this important point. We have revised this section. The role of neutrophils in Mtb dissemination is an emerging area of research, with growing evidence suggesting that these cells contribute to the spread of Mtb beyond the lungs (Hult et al., 2021). We highlight that the observed correlation could be speculative at this juncture.

Figure 4, 5: IL-17 neutralization alone has no effect on CFU in the lungs of Mtb-infected mice. While the combination of IL-17 neutralization and celecoxib has a very modest effect on CFU, the mechanism behind this observation is unclear. Further, the experiment shown has only 3 mice per group and it is unclear whether this (or any other) mouse experiment was repeated.

For Fig. 4, the experiment was done with 3 mice/group. The IFN KO mice were used to help identify the mechanism. IL-17 neutralisation or Celecoxib treatment alone did not have any significant effect on the bacterial burden (in lungs or isolated PMNs). However, it did show a significant effect on the number of PMNs recruited. Combination of IL-17 neutralisation and celecoxib led to about a one-log decrease in CFU, which is significant.

For Fig. 5, we used SR2211 instead of anti-IL-17 Ab for the experiment. This experiment had WT mice and 5 animals/group. Here, celecoxib and SR2211 alone showed a significant decline in PMN-resident Mtb pool as well as spleen burden. Only in the lungs, the impact of SR2211 alone was not significant.

Figure 6: The decreases in CFU correlate with a decrease in neutrophils; nothing connects this to neutrophil production of IL-17.

We now show quantification of observation in Fig. 5I, where in the WT mice, treatment with Celecoxib reduces the frequency of IL-17-producing Ly6G+ cells. In the revised manuscript, we also show direct evidence of SR2211 activity on BCG vaccine efficacy, which causes a significant decline in the Mtb burden in whole lung or in the isolated PMNs.

Figure 7. The Human data shows that elevated neutrophil levels and elevated IL-17 levels are associated with treatment failure in TB patients. This is expected, and does not

The literature lacks consensus in terms of a protective or pathological role of IL-17 in TB. Therefore, it was not expected to see higher IL-17 in patients who experienced relapse, death, or failed treatment outcomes. We do not have evidence from human subjects whether neutrophil derived IL-17 has a similar pathological role as observed in mice. However, higher IL-17 in failed outcome cases confirm the central theme that IL-17 is pathological in both human and mouse models.

**Reviewer #2 (Recommendations for the authors):**
(1) Survival of IFN-γ-/- Mice: The survival of IFN-γ-/- mice up to 100 days following a challenge with ~100 CFU of H37Rv is quite unusual. Have the authors checked PDIM expression in their Mtb strain, given that several studies report earlier mortality in these mice?

As shown in Fig. 4F, H37Rv-infected IFN-γ⁻/⁻ mice survived up to a little over 80 days. These figures are not unusual in the light of the following:

(1) In one study, IFNγ⁻/⁻ survived for about 40 days when the hypervirulent Mtb strain was used to infect these mice at 100-200 CFU using nose-only aerosol exposure (Nandi and Behar, 2011)

(2) In yet another study, IFNγ⁻/⁻ mice survived for ~50 days, however, they used H37Rv at 1-3x10^5^ CFU to infect through intravenous injection (Kawakami et al., 2004)

Thus, compared with the above observations, where IFN-γ^-/-^ mice survived for maximum 50 days due to hypervirulent infection or a very high dose infection, infection with H37Rv at ~100 CFU through the aerosol route and surviving for ~80 days is not unusual. The H37Rv cultures used in our study are always animal-passaged to ensure PDIM integrity.

(2) Granuloma Scoring: The granuloma scores appear to represent the percentage of lesion area. Please clarify and, if necessary, amend this in the manuscript.

The granuloma score is based on the calculation of the number of granulomatous infiltration and their severity. These are not % lesion area. We have added this detail in the revised manuscript.

(3) Pathology Comparison in Figures 4F and 4G: Does the pathology shown in Figure 4G correspond to the same groups as in Figure 4F? The celecoxib group in Figure 4F and the WT group in Figure 4G seem to be missing. Please clarify.

Figures 4F and 4G depict two independent experiments. For the time-to-death experiment, we had to leave the animals. The rest of the panels in Fig. 4 represent animals from the same experiment.

(4) Effect of Celecoxib on Ly6G+ Cells: The authors demonstrated that celecoxib treatment reduces Ly6G+ cells and IL-17-producing Ly6G+ cells. Do Ly6G+ cells express EP2/EP4 receptors? Alternatively, could the reduction in IL-17-producing Ly6G+ cells be due to an improved bactericidal response in other innate cells? The authors should discuss this possibility.

Yes, Ly6G^⁺^ granulocytes express EP2/EP4 receptors (Lavoie et al., 2024), which mediate PGE₂ signaling. Prostaglandin E_₂_ (PGE_₂_) is known to regulate neutrophil function and can enhance IL-17 production in various immune cells (Napolitani et al., 2009). However, the expression and functional role of EP2/EP4 receptors specifically on Ly6G^⁺^ granulocytes in the context of Mtb infection require further investigation.

The alternate suggestion by the reviewer that the reduction in IL-17-producing Ly6G^⁺^ cells following celecoxib treatment could be attributed to an improved bactericidal response in other innate immune cells is attractive. While we did not experimentally rule out this possibility, since reduced IL-17 invariably associated with reduced neutrophil-resident Mtb population, a cell-autonomous mechanism operational in Ly6G+ granulocytes is a highly likely mechanism.

(5) Culture Conditions: The methods section indicates that bacteria were cultured in 7H9+ADC. Is there a specific reason why the Oleic acid supplement was not added, given that standard Mtb culture conditions typically use 7H9+OADC supplements? Please comment on this choice.

It is a standard microbiological experimental procedure to use 7H9+ADC for broth culture, while 7H11+OADC for solid culture. Compared to broth culture, solid media are usually more stressful for bacteria because of hypoxia inside the growing colonies. Therefore, the media used are enriched in casein hydrolysate (like 7H11) and oleic acid (OADC).

**Reviewer #3 (Recommendations for the authors):**
Major suggestion: To really determine the role of neutrophil IL17 will require depletion studies and chimera experiments. These are clearly a major undertaking. I believe making significant re-writes to alter the conclusions or reanalyze any data to determine the role of nonhematopoietic and hematopoietic cells in IL17 is needed. If the conclusions are left as is, further experimentation is needed to fully support those conclusions.

Thank you for the suggestion. We have embarked on the specific deletion studies; however, as mentioned, this is a major undertaking and will take time. As suggested, we have discussed the results in accordance with the strength of evidence currently provided.

Eum, S.Y., J.H. Kong, M.S. Hong, Y.J. Lee, J.H. Kim, S.H. Hwang, S.N. Cho, L.E. Via, and C.E. Barry, 3rd. 2010. Neutrophils are the predominant infected phagocyGc cells in the airways of paGents with acGve pulmonary TB. Chest 137:122-128.

Gonzalez-Orozco, M., R.E. Barbosa-Cobos, P. Santana-Sanchez, L. Becerril-Mendoza, L. Limon-

Camacho, A.I. Juarez-Estrada, G.E. Lugo-Zamudio, J. Moreno-Rodriguez, and V. OrGzNavarrete. 2019. Endogenous sGmulaGon is responsible for the high frequency of IL-17Aproducing neutrophils in paGents with rheumatoid arthriGs. Allergy Asthma Clin Immunol 15:44.

References

Hult, C., J.T. Mala, H.P. Gideon, J.J. Linderman, and D.E. Kirschner. 2021. Neutrophil Dynamics Affect *Mycobacterium tuberculosis* Granuloma Outcomes and DisseminaGon. Front Immunol 12:712457.

Kawakami, K., Y. Kinjo, K. Uezu, K. Miyagi, T. Kinjo, S. Yara, Y. Koguchi, A. Miyazato, K. Shibuya, Y. Iwakura, K. Takeda, S. Akira, and A. Saito. 2004. Interferon-gamma producGon and host protecGve response against *Mycobacterium tuberculosis* in mice lacking both IL-12p40 and IL-18. Microbes Infect 6:339-349.

Lavoie, J.C., M. Simard, H. Kalkan, V. Rakotoarivelo, S. Huot, V. Di Marzo, A. Cote, M. Pouliot, and N. Flamand. 2024. Pharmacological evidence that the inhibitory effects of prostaglandin E2 are mediated by the EP2 and EP4 receptors in human neutrophils. J Leukoc Biol 115:1183-1189.

Lee, J., S. Boyce, J. Powers, C. Baer, C.M. Sasse, and S.M. Behar. 2020. CD11cHi monocyte-derived macrophages are a major cellular compartment infected by *Mycobacterium tuberculosis*. PLoS Pathog 16:e1008621.

Li, L., L. Huang, A.L. Vergis, H. Ye, A. Bajwa, V. Narayan, R.M. Strieter, D.L. Rosin, and M.D. Okusa. 2010. IL-17 produced by neutrophils regulates IFN-gamma-mediated neutrophil migraGon in mouse kidney ischemia-reperfusion injury. J Clin Invest 120:331-342.

Lovewell, R.R., C.E. Baer, B.B. Mishra, C.M. Smith, and C.M. Sasse. 2021. Granulocytes act as a niche for *Mycobacterium tuberculosis* growth. Mucosal Immunol 14:229-241.

Nandi, B., and S.M. Behar. 2011. RegulaGon of neutrophils by interferon-gamma limits lung inflammaGon during tuberculosis infecGon. The Journal of experimental medicine 208:22512262.

Napolitani, G., E.V. Acosta-Rodriguez, A. Lanzavecchia, and F. Sallusto. 2009. Prostaglandin E2 enhances Th17 responses via modulaGon of IL-17 and IFN-gamma producGon by memory CD4+ T cells. Eur J Immunol 39:1301-1312.

Ramirez-Velazquez, C., E.C. CasGllo, L. Guido-Bayardo, and V. OrGz-Navarrete. 2013. IL-17-producing peripheral blood CD177+ neutrophils increase in allergic asthmaGc subjects. Allergy Asthma Clin Immunol 9:23.

Sadikot, R.T., H. Zeng, A.C. Azim, M. Joo, S.K. Dey, R.M. Breyer, R.S. Peebles, T.S. Blackwell, and J.W. Christman. 2007. Bacterial clearance of *Pseudomonas aeruginosa* is enhanced by the inhibiGon of COX-2. Eur J Immunol 37:1001-1009.

Zheng, W., I.C. Chang, J. Limberis, J.M. Budzik, B.S. Zha, Z. Howard, L. Chen, and J.D. Ernst. 2023. *Mycobacterium tuberculosis* resides in lysosome-poor monocyte-derived lung cells during chronic infecGon. bioRxiv

Zheng, W., I.C. Chang, J. Limberis, J.M. Budzik, B.S. Zha, Z. Howard, L. Chen, and J.D. Ernst. 2024. *Mycobacterium tuberculosis* resides in lysosome-poor monocyte-derived lung cells during chronic infecGon. PLoS Pathog 20:e1012205.